# HYPER: A FOUNDATION MODEL FOR INDUCTIVE LINK PREDICTION WITH KNOWLEDGE HYPERGRAPHS

**Xingyue Huang**
University of Oxford

**Mikhail Galkin**
Google Research

**Michael M. Bronstein**
University of Oxford & AITHYRA

**İsmail İlkan Ceylan**
TU Wien & AITHYRA & University of Oxford

## ABSTRACT

Inductive link prediction with knowledge hypergraphs is the task of predicting missing hyperedges involving completely *novel entities* (i.e., nodes unseen during training). Existing methods for inductive link prediction with knowledge hypergraphs assume a fixed relational vocabulary and, as a result, cannot generalize to knowledge hypergraphs with *novel relation types* (i.e., relations unseen during training). Inspired by knowledge graph foundation models, we propose HYPER as a foundation model for link prediction, which can generalize to *any knowledge hypergraph*, including novel entities and novel relations. Importantly, HYPER can learn and transfer across different relation types of *varying arities*, by encoding the entities of each hyperedge along with their respective positions in the hyperedge. To evaluate HYPER, we construct 16 new inductive datasets from existing knowledge hypergraphs, covering a diverse range of relation types of varying arities. Empirically, HYPER consistently outperforms all existing methods in both node-only and node-and-relation inductive settings, showing strong generalization to unseen, higher-arity relational structures.

## 1 INTRODUCTION

Generalizing knowledge graphs with relations between *any* number of nodes, knowledge hypergraphs offer flexible means of storing, processing, and managing *relational data*. Knowledge hypergraphs can encode rich relationships between entities; e.g., consider a relationship between *four* entities: "Bengio has a research project on topic ClimateAI in Montreal funded by CIFAR". This relational information can be represented in a knowledge hypergraph (see Figure 1) via an (ordered) hyperedge Research(Bengio, ClimateAI, Montreal, CIFAR), where Research represents a relation of arity *four*.

The generality of knowledge hypergraphs motivated a body of work for machine learning with knowledge hypergraphs (Guan et al., 2021; Fatemi et al., 2020; Yadati, 2020; Zhou et al., 2023b; Huang et al., 2025b). One of the most prominent learning tasks is inductive link prediction with knowledge hypergraphs, where the goal is to predict missing hyperedges involving completely *novel*

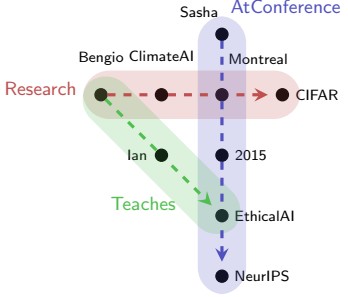

Figure 1: A knowledge hypergraph with three hyperedges over distinct relation types.

*entities* (Yadati, 2020; Zhou et al., 2023b; Huang et al., 2025b). The main shortcoming of existing methods for inductive link prediction with knowledge hypergraphs is that they cannot generalize to knowledge hypergraphs with *novel relation types*. This constitutes the main motivation of our work: *Can we design an effective model architecture for inductive link prediction with knowledge hypergraphs, where the predictions can involve both novel entities and novel relations?*

**Example.** Consider the knowledge hypergraphs depicted in Figure 2: The training hypergraph $G_{train}$ is over the relations Research, Teaches, and AtConference, while the inference graph $G_{inf}$ is over the novel relations Trading, Sells, and AtFair. The task is to predict missing links such as

Sells(Samsung, Best Buy, Q60D TV) in $G_{inf}$. Ideally, the model should learn relation invariants that map Teaches $\mapsto$ Sells, Research $\mapsto$ Trading, and AtConference $\mapsto$ AtFair, as these relation types play analogous structural roles in their respective graphs, even though their labels and entities are entirely different.

**Approach.** In essence, our study builds on the success of knowledge graph foundation models (KGFMs) (Galkin et al., 2024; Mao et al., 2024), which have shown remarkable performance in link prediction tasks involving both novel entities and novel relations. However, KGFMs can only perform link prediction using *binary relations*, which raises the question of how to translate the success of KGFMs to fully relational data. To this end, we propose HYPER, a class of knowledge hypergraph foundation models for inductive link prediction, which can generalize to any knowledge hypergraph. The fundamental idea behind our approach is to learn properties of relations that are transferable between different types

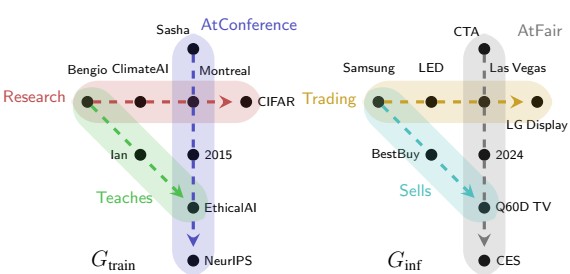

Figure 2: A model is trained on relations like Research, Teaches, and AtConference, and is expected to generalize to structurally similar relations TradingDeal, Sells, and AtBusinessFair at test time.

of relations of varying arity. Consider, for example, the two hyperedges (from Figure 2):

$$\text{AtConference}(\text{Sasha}, \underline{\text{Montreal}}, 2015, \text{EthicalAI}, \text{NeurIPS}),$$
$$\text{Research}(\text{Bengio}, \text{ClimateAI}, \underline{\text{Montreal}}, \text{CIFAR}),$$

which "intersect" with each other. The entity Montreal appears in the *second* position of the first hyperedge and in the *third* position of the second hyperedge. Such (pairwise) interactions between relations can be viewed as *fundamental relations* to learn from: any model learning from relations between relations can transfer this knowledge to novel relation types that have similar interactions.

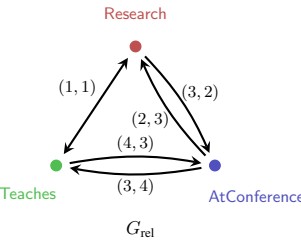

Figure 3: The relation graph $G_{rel}$ corresponding to the knowledge hypergraph $G_{train}$.

Furthermore, we can encode such relations between relations in a separate relation graph, which can be used to learn from. We illustrate this on our running example in Figure 3, where the relations appear as nodes; the interactions between relations as edges; and finally, the positions of the interactions as edge weights. In our setting, a directed edge from relation $r_1$ to $r_2$ with edge label $(i, j)$ indicates that *"The i-th position of $r_1$ and the j-th position of $r_2$ intersect in G"*, which captures a fundamental interaction between $r_1$ and $r_2$. Critically, however, there is no upper bound on the number of such possible interactions. While there are at most $m \times n$ interactions between an $m$-ary relation and an $n$-ary relation, we cannot impose any bound on the arity of the relations since then the model would not generalize to all knowledge hypergraphs.

**Contributions.** Our main contributions can be summarized as follows:

- To the best of our knowledge, HYPER is the first foundation model that allows zero-shot generalization to knowledge hypergraphs of *arbitrary* arity with *novel nodes* and *novel relations* at test time.

- We evaluate HYPER on 3 existing benchmark datasets and additionally on 16 new benchmark datasets with varying proportions of test-time tuples involving unseen relations. HYPER consistently outperforms existing hypergraph baselines trained end-to-end, particularly when the proportion of new relations is high.

- To assess the performance of KGFMs on hypergraphs, we reify the knowledge hypergraphs into KGs and apply KGFMs on them. Remarkably, HYPER, trained on only 2 hypergraphs and 3KGs, consistently outperforms the popular KGFM model ULTRA trained on 50 KGs.

- We conduct an empirical investigation over the positional interaction encoding scheme within HYPER, demonstrating the critical role of encoding choices.

## 2 RELATED WORK

**Link Prediction with Knowledge Graphs.**
Early knowledge graph embedding methods
(Bordes et al., 2013; Sun et al., 2019; Trouillon et al., 2016; Balazevic et al., 2019; Abboud et al., 2020) are limited to the *transductive* setup: these methods do not generalize to unseen entities or to unseen relations. Multi-relational graph neural networks (GNNs) such as RGCN (Schlichtkrull et al., 2018) and CompGCN (Vashishth et al., 2020) similarly rely on stored entity embeddings, remaining inherently transductive.

Table 1: Methods' ability to handle high-arity relations (**High-arity**) and inductively generalize to unseen entities (**Ind.** $e$) and relations (**Ind.** $r$).

| Methods | High-arity | Ind. $e$ | Ind. $r$ |
|---|---|---|---|
| HypE, BoxE | ✓ | ✗ | ✗ |
| NBFNet, A*Net | ✗ | ✓ | ✗ |
| G-MPNN, HCNet | ✓ | ✓ | ✗ |
| ULTRA, KG-ICL | ✗ | ✓ | ✓ |
| HYPER | ✓ | ✓ | ✓ |

To overcome these limitations, Teru et al. (2020) introduced GraIL, a pioneering method enabling *node-inductive* link prediction, which is later shown to be a form of the labeling trick (Zhang et al., 2021). Subsequently, architectures such as NBFNet (Zhu et al., 2021), A*Net (Zhu et al., 2023), RED-GNN (Zhang & Yao, 2022), and AdaProp (Zhang et al., 2023) leveraged conditional message passing, significantly enhancing expressivity and performance (Huang et al., 2023). However, these methods are *not* inductive on relations, as they assume a fixed relational vocabulary. KGFMs are specifically tailored for inductive predictions on both unseen nodes and relations. InGram (Lee et al., 2023), RAILD (Gesese et al., 2022), and ULTRA (Galkin et al., 2024) introduced new KGFM frameworks. Following these, TRIX (Zhang et al., 2024) introduced recursive updating of entity and relation embeddings with provably improved expressiveness over ULTRA. KG-ICL (Cui et al., 2024) employed in-context learning with unified tokenization for entities and relations. Additionally, double-equivariant GNNs, like ISDEA (Gao et al., 2023) and MTDEA (Zhou et al., 2023a), emphasized relational equivariance, enhancing robustness to unseen relations. Huang et al. (2025a) proposed MOTIF as a general KGFM framework and formally studied the expressive power of KGFMs. All of these methods are confined to KGs with binary relations, and they do not naturally apply to higher-arity relations, as shown in Table 1.

**Link Prediction with Knowledge Hypergraphs.** Knowledge hypergraphs generalize traditional KGs to handle higher-arity relational data. Initial researches such as HypE (Fatemi et al., 2020) and BoxE (Abboud et al., 2020) leveraged shallow embedding models adapted from KG embedding frameworks. Later approaches extended graph neural networks to knowledge hypergraphs. G-MPNN (Yadati, 2020) and RD-MPNNs (Zhou et al., 2023b) introduced relational message passing mechanisms explicitly designed for hypergraph settings, incorporating positional entity information critical for high-arity relations. Huang et al. (2025b) proposed HCNets as a conditional message-passing approach tailored for inductive hypergraph link prediction and conducted an expressivity analysis. While these methods can handle knowledge hypergraphs, they are *not* inductive on relations: none of these methods can generalize to unseen relations (shown in Table 1). Our work on HYPER builds on these foundations by combining the strengths of conditional message passing on knowledge hypergraphs with the powerful inductive generalization techniques explored in recent KGFMs (Galkin et al., 2024; Lee et al., 2023; Huang et al., 2025a) to effectively generalize to knowledge hypergraphs within unseen nodes and relations.

**Foundation Models on Hypergraphs.** Existing foundation models on hypergraphs are tailored to text-attributed hypergraphs. HyperBERT (Bazaga et al., 2024) integrates pretrained language models with hypergraph convolution for node classification, while HyperGene (Du et al., 2021) and SPHH (Abubaker et al., 2023) propose self-supervised objectives tailored to local and global hypergraph structures. More recent works such as Hyper-FM (Feng et al., 2025) and IHP (Yang et al., 2024) introduce multi-domain pretraining and instruction-guided adaptation, respectively, marking the first steps toward generalizable hypergraph models. These methods rely heavily on text attributes for generalization and are predominantly tailored to node classification tasks; they do not support link prediction over knowledge hypergraphs with unseen relations at test time.

## 3 PRELIMINARIES

**Knowledge Hypergraphs.** A *knowledge hypergraph* $G = (V, E, R)$ consists of a set of nodes $V$, hyperedges $E$ (i.e., facts) of the form $e = r(u_1, \ldots, u_k)$, where $r \in R$ is a relation type,

and $u_i \in V$, $1 \leq i \leq k$, are nodes. The arity of a relation $r$ is given by $k = \texttt{ar}(r)$, where $\texttt{ar} : R \mapsto \mathbb{N}_{>0}$. For an hyperedge $e$, $\rho(e)$ denotes its relation, and $e(i)$ denotes the node at the $i$-th position of $e$. We refer to the knowledge hypergraph with all edges having arity of exactly 2 as a *knowledge graph*. The set of edge-position pairs associated with a node $v$ is defined as: $E(v) = \{(e, i) \mid e(i) = v, e \in E, 1 \leq i \leq \texttt{ar}(\rho(e))\}$. The positional neighborhood of a hyperedge $e$ with respect to a position $i$ is: $\mathcal{N}_i(e) = \{(e(j), j) \mid j \neq i, 1 \leq j \leq \texttt{ar}(\rho(e))\}$.

**Link Prediction on Hyperedges.** Given a knowledge hypergraph $G = (V, E, R)$ and a query $q(u_1, \ldots, u_{t-1}, ?, u_{t+1} \ldots, u_k)$, the *link prediction* task involves scoring all possible hyperedges formed by replacing the placeholder '?' with each node $v \in V$. We denote a $k$-tuple of nodes by $\mathbf{u} = (u_1, \ldots, u_k)$ and the tuple excluding position $t$ by $\tilde{\mathbf{u}} = (u_1, \ldots, u_{t-1}, u_{t+1}, \ldots, u_k)$. Thus, we represent a query succinctly as $\mathbf{q} = (q, \tilde{\mathbf{u}}, t)$. In the *fully-inductive setting* for link prediction (i.e., node and relation-inductive link prediction), the goal is to answer queries of the form $\mathbf{q} = (q, \tilde{\mathbf{u}}, t)$ on an inference hypergraph $G_{\text{inf}} = (V_{\text{inf}}, E_{\text{inf}}, R_{\text{inf}})$, where both the entity set $V_{\text{inf}}$ and the relation set $R_{\text{inf}}$ are entirely disjoint from those seen during training. The model is trained on a separate training knowledge hypergraph $G_{\text{train}} = (V_{\text{train}}, E_{\text{train}}, R_{\text{train}})$, with $V_{\text{train}} \cap V_{\text{inf}} = \emptyset$ and $R_{\text{train}} \cap R_{\text{inf}} = \emptyset$, and must learn transferable representations that generalize across both novel entities and unseen relation types of arbitrary arity. At inference time, each hyperedge $e = r(u_1, \ldots, u_k) \in E_{\text{inf}}$ corresponds to a fact involving a relation $r \in R_{\text{inf}}$, and queries involve predicting a missing node at position $t$ within such a tuple, using the surrounding nodes $\tilde{\mathbf{u}}$ and relation $q = \rho(e)$. The model must score candidate completions $q(u_1, \ldots, u_{t-1}, v, u_{t+1}, \ldots, u_k)$ for each $v \in V_{\text{inf}}$.

**Reification.** To apply the models designed for KGs on knowledge hypergraphs, we transform an input knowledge hypergraph $G = (V, E, R)$ into a KG via a *reification* process, similar to the one proposed in Fatemi et al. (2020)[1]. Specifically, for each hyperedge $r(u_1, \ldots, u_k) \in E$, we introduce a node $\texttt{edge\_id} \notin V$ in the KG to represent the hyperedge itself and add $k$ binary edges

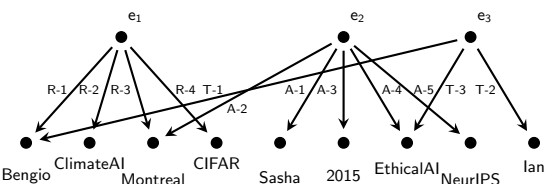

Figure 4: Reified KG corresponding to the knowledge hypergraph $G_{\text{train}}$ from Fig 1. R-i abbreviates Research-i, similarly for A as AtConference, and T as Teaches.

$r\text{-}1\big(\texttt{edge\_id}, u_1\big)$, $r\text{-}2\big(\texttt{edge\_id}, u_2\big)$, $\ldots$, $r\text{-}k\big(\texttt{edge\_id}, u_k\big)$, one for each position. Note that each new edge encodes a *position-specific* relation such as Research-3 or AtConference-5. For instance, Figure 4 shows the reified KG of our running example in Figure 1. This reification procedure encodes the full higher-order structure of the original knowledge hypergraph into a KG.

**Link Prediction over Reified Knowledge Hypergraphs.** Given a high-arity query of the form $q(u_1, \ldots, u_{t-1}, ?, u_{t+1}, \ldots, u_k)$ over the original knowledge hypergraph, we perform link prediction in the reified KG by encoding the query as a subgraph which is used to augment the testing knowledge graph. Concretely, we add a new node $\texttt{edge\_id}$ and binary triples $q_i(\texttt{edge\_id}, u_i)$ for all $i \neq t$. The prediction task is then reduced to a standard tail prediction problem: ranking all candidate entities $v \in V$ for the fact $q_t(\texttt{edge\_id}, v)$. We evaluate the model performance using standard ranking metrics over the original entity vocabulary. We use superscript ($\ddagger$) to denote models evaluated under this regime.

## 4 HYPER: A KNOWLEDGE HYPERGRAPH FOUNDATION MODEL

We now present HYPER, a general framework for learning foundation models over knowledge hypergraphs. Given a knowledge hypergraph $G = (V, E, R)$ and a query $\boldsymbol{q} = (q, \tilde{\boldsymbol{u}}, t)$, HYPER computes link prediction scores through the following steps:

1. **Relation encoder:** Relations are encoded in three steps:

---

[1] We also include alternative ways for reification in Appendix G.3.

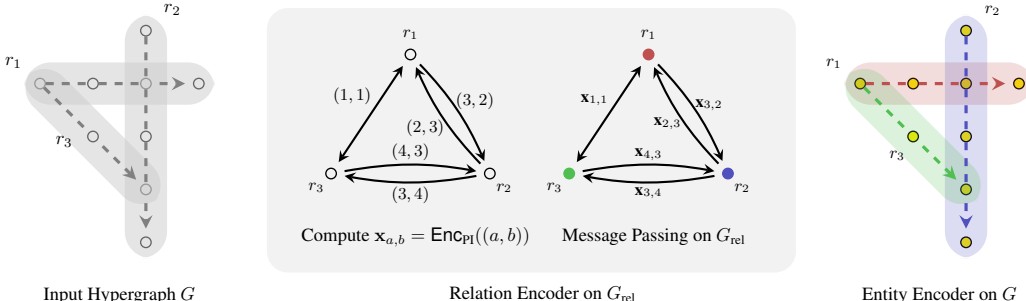

Figure 5: Overall framework of HYPER. HYPER first constructs a relation graph $G_{\text{rel}}$ based on the observed positional interactions between the relations. $\mathsf{Enc}_{\text{PI}}$ then computes embeddings for each position pair, which are refined via message passing over $G_{\text{rel}}$. The resulting relation representations are then used for message passing over the original knowledge hypergraph $G$ (shown in color).

(a) **Relation graph:** Build a relation graph $G_{\text{rel}}$ where each node corresponds to a relation $r \in R$, and edges capture observed positional interactions between relations.

(b) **Encoding positional interactions:** Use an encoder $\mathsf{Enc}_{\text{PI}}$ to embed each interacting position pair $(a, b)$ from $G_{\text{rel}}$ into fundamental relation representations.

(c) **Encoding the relations:** Perform conditional message passing over $G_{\text{rel}}$ using fundamental relation representations to obtain relation embeddings for all $r \in R$.

2. **Entity encoder:** Use learned relation representations to conduct conditional message passing over the original knowledge hypergraph $G$ and obtain link probability via decoder $\mathsf{Dec}$.

The overall framework is illustrated in Figure 5, and the detailed architecture is presented in Appendix G. We now describe each component.

**Relation graph.** Given a knowledge hypergraph $G = (V, E, R)$, we construct the relation graph $G_{\text{rel}} = (V_{\text{rel}}, E_{\text{rel}}, R_{\text{rel}})$. The set of nodes is given as $V_{\text{rel}} = R$, i.e., each node in $G_{\text{rel}}$ corresponds to a relation type in the knowledge hypergraph $G$. The relation types $R_{\text{rel}}$ are defined as all ordered pairs $(a, b)$ for $\{1 \leq a, b \leq k_{\max}\}$, where $k_{\max} = \max\{\mathtt{ar}(r) \mid r \in R\}$ denotes the maximum arity among the observed relations. The edge set $E_{\text{rel}}$ captures *positional interactions* between relation types: for each pair of hyperedges $e_1, e_2 \in E$ with relation types $r_1 = \rho(e_1)$ and $r_2 = \rho(e_2)$, if there exists a shared entity $v$ appearing in position $i$ in $e_1$ and position $j$ in $e_2$, we add a directed edge $(r_1, r_2)$ with relation type $(i, j)$ to $E_{\text{rel}}$. These positional interactions can be computed efficiently via sparse matrix multiplication (see Appendix B) and are invariant over the renaming of relations.

**Encoding positional interactions.** Unlike knowledge graphs, where each fact involves two entities and naturally leads to four types of fundamental relations (head-to-head, head-to-tail, tail-to-head, and tail-to-tail) as introduced in Galkin et al. (2024), knowledge hypergraphs allow facts with arbitrary arity. This introduces a key challenge: *How to build a foundation model that can adapt to unseen knowledge hypergraphs with varying and arbitrarily large arities?*

The natural extension of the concept of fundamental relations from KGs to knowledge hypergraphs results in $mn$ types of positional interactions between an hyperedge of arity $m$ and an hyperedge of arity $n$. Each of such interaction is characterized by a pair $(a, b)$, where $a$ and $b$ denote the entity positions involved in the relation. As a consequence, a foundation model for knowledge hypergraphs must be capable of encoding positional interactions in a way that generalizes across different arities.

A naive solution would be to associate a separate embedding to each $(a, b)$ pair. However, such an approach does not generalize to unseen arities, as it would require pre-training embeddings for all possible $(a, b)$ combinations. To address this, we propose a *shared, compositional position interaction encoding scheme*. Specifically, given a positional interaction labeled $(a, b)$, we define a positional interaction encoder $\mathsf{Enc}_{\text{PI}} : \mathbb{N}_{>0} \times \mathbb{N}_{>0} \to \mathbb{R}^d$, which maps a pair of argument positions to a dense vector representation of $d$ dimensions. To be effective in inductive settings, we require the encoder $\mathsf{Enc}_{\text{PI}}$ to satisfy the following requirements:

1. **Extrapolation.** The encoder should generalize to unseen positions and combinations, allowing the model to operate on arities and interaction patterns not present during training.

2. **Injectivity.** Distinct position pairs $(a, b)$ and $(a', b')$ should map to distinct embeddings to preserve the identifiability of positional interactions:

$$\forall a, b, a', b' \in \mathbb{N}_{>0}, (a, b) \neq (a', b') \implies \mathsf{Enc}_{\mathrm{PI}}((a, b)) \neq \mathsf{Enc}_{\mathrm{PI}}((a', b')).$$

In practice, we implement $\mathsf{Enc}_{\mathrm{PI}}$ as a two-layer multilayer perceptron (MLP) over concatenated sinusoidal encodings of the input positions. Let $\boldsymbol{p}_a, \boldsymbol{p}_b \in \mathbb{R}^d$ denote the sinusoidal positional encodings of positions $a$ and $b$, respectively. Then, the embedding corresponding to the interaction $(a, b)$ is computed as $\mathbf{x}_{a,b} = \mathrm{MLP}([\boldsymbol{p}_a \,\|\, \boldsymbol{p}_b])$, where MLP denotes a shared two-layer feedforward network with ReLU activations. This produces a dense embedding that captures the interaction between the two positions. Theoretically, we show that our chosen $\mathsf{Enc}_{\mathrm{PI}}$ satisfies these properties and is also locally smooth. (See Appendix C for formal theorems and proofs.)

**Theorem 4.1** (Informal). *There exists a set of parameter for* HYPER *such that* $\mathsf{Enc}_{\mathrm{PI}}$ *is injective, has a bounded range, and is Lipschitz (and hence locally smooth).*

Empirically, we find that this instantiation of $\mathsf{Enc}_{\mathrm{PI}}$ also enables strong generalization across knowledge hypergraphs with varying arities and relational structures. When applied to knowledge graphs, our method recovers standard encoding patterns employed in many KGFMs (Galkin et al., 2024; Lee et al., 2023; Zhang et al., 2024; Huang et al., 2025a). In particular, head-to-tail, head-to-head, tail-to-tail, and tail-to-head interactions correspond to $\mathsf{Enc}_{\mathrm{PI}}((1, 2))$, $\mathsf{Enc}_{\mathrm{PI}}((1, 1))$, $\mathsf{Enc}_{\mathrm{PI}}((2, 2))$, and $\mathsf{Enc}_{\mathrm{PI}}((2, 1))$, respectively.

**Encoding the relations.** HYPER uses *Hypergraph Conditional Networks* (HCNets) (Huang et al., 2025b) to encode relations for its strong inductive performance, support for bidirectional message passing, and easy extensibility to higher-order relational patterns (Huang et al., 2025a). HCNets produce query-conditioned representations by aggregating messages from neighboring edges with relation and position information. Here, we take $\mathsf{Enc}_{\mathrm{PI}}((a, b))$ as the computed messages for each typed edges when message-passing over relation graph with positional encoding $(a, b)$.

**Entity encoder.** Similarly to how we encode the relation, HYPER uses a variant of HCNet to encode the entities. In the context of the entity encoder, we apply a separate HCNet over the original knowledge hypergraph $G = (V, E, R)$. Each node $v \in V$ aggregates from its incident hyperedges by first taking the relation embeddings $\boldsymbol{h}_{r|\boldsymbol{q}}^{(T)}$ obtained from the relation encoder as the messages for each typed hyperedges and then transformed by a layer-specific MLP. The resulting HYPER instances will preserve equivariance over nodes and relations. (See Appendix C for proof.)

## 5 EXPERIMENTS

In this section, we aim to evaluate the generalization and effectiveness of HYPER across inductive link prediction tasks on both knowledge hypergraphs and knowledge graphs. We focus on answering the following questions:

**Q1:** How well does HYPER generalize to unseen entities and relation types?

**Q2:** How does HYPER handle varying proportions of unseen relations in the test set?

**Q3:** How does HYPER compare to KGFMs on reified knowledge hypergraphs?

**Q4:** What is the impact of different variants of pretraining mix on HYPER?

**Q5:** How does the encoding of positional information impact the model's ability to generalize?

**Q6:** How well does HYPER perform on standard knowledge graphs (see Appendix E)?

**Q7:** What are computational complexity and empirical scalability of HYPER (see Appendix F)?

### 5.1 EXPERIMENTAL SETUPS

**Models.** We evaluate models using the datasets summarized in Table 13. As a supervised learning baseline, we include G-MPNN (Yadati, 2020) and HCNet as node-inductive methods on knowledge

Table 2: MRR results on node and relation inductive knowledge hypergraph datasets. Superscript ‡ means the model is applied over the reification of hypergraphs.

| Method | JF | | | | MFB | | | | WP | | | | WD | | | |
|---|---|---|---|---|---|---|---|---|---|---|---|---|---|---|---|---|
| | 25 | 50 | 75 | 100 | 25 | 50 | 75 | 100 | 25 | 50 | 75 | 100 | 25 | 50 | 75 | 100 |
| **End-to-End Inference** | | | | | | | | | | | | | | | | |
| G-MPNN | 0.006 | 0.003 | 0.001 | 0.002 | 0.002 | 0.004 | 0.007 | 0.003 | 0.005 | 0.002 | 0.001 | 0.000 | 0.001 | 0.001 | 0.001 | 0.001 |
| HCNet | 0.011 | 0.009 | 0.069 | 0.028 | 0.033 | 0.026 | 0.016 | 0.082 | 0.104 | 0.050 | 0.019 | 0.003 | 0.086 | 0.043 | 0.015 | 0.007 |
| HYPER | **0.202** | **0.468** | **0.207** | **0.198** | **0.332** | **0.200** | **0.135** | **0.222** | **0.159** | **0.143** | **0.139** | **0.202** | **0.215** | **0.205** | **0.172** | **0.205** |
| **Zero-shot Inference** | | | | | | | | | | | | | | | | |
| ULTRA‡(3KG) | 0.103 | 0.437 | 0.168 | 0.144 | 0.255 | 0.235 | 0.154 | 0.277 | 0.039 | 0.077 | 0.073 | 0.078 | 0.117 | 0.155 | 0.116 | 0.161 |
| HYPER(3KG) | 0.148 | 0.297 | 0.112 | 0.130 | 0.248 | 0.191 | 0.039 | 0.276 | **0.143** | 0.147 | 0.186 | 0.221 | 0.167 | 0.158 | 0.123 | 0.146 |
| ULTRA‡(4KG) | 0.011 | 0.298 | 0.042 | 0.082 | 0.217 | 0.170 | 0.043 | 0.135 | 0.009 | 0.006 | 0.006 | 0.004 | 0.027 | 0.069 | 0.063 | 0.065 |
| HYPER(4KG) | 0.109 | 0.065 | 0.128 | 0.087 | 0.116 | 0.117 | 0.089 | 0.148 | 0.074 | 0.212 | 0.175 | 0.180 | 0.148 | 0.111 | 0.150 | 0.255 |
| ULTRA‡(50KG) | 0.001 | 0.096 | 0.010 | 0.001 | 0.225 | 0.083 | 0.001 | 0.190 | 0.006 | 0.009 | 0.009 | 0.004 | 0.008 | 0.001 | 0.001 | 0.001 |
| HYPER(50KG) | 0.056 | 0.294 | 0.084 | 0.111 | 0.122 | 0.156 | 0.126 | 0.148 | 0.067 | 0.198 | 0.191 | 0.155 | 0.073 | 0.055 | 0.130 | 0.088 |
| ULTRA‡(4HG) | 0.154 | 0.442 | 0.175 | 0.170 | 0.338 | 0.236 | 0.129 | 0.280 | 0.052 | 0.091 | 0.089 | 0.089 | 0.076 | 0.175 | 0.052 | 0.136 |
| HYPER(4HG) | 0.187 | 0.377 | 0.188 | **0.181** | 0.349 | 0.244 | 0.139 | 0.278 | 0.075 | 0.068 | 0.086 | 0.168 | 0.087 | 0.158 | 0.057 | 0.165 |
| ULTRA‡(3KG+2HG) | 0.209 | 0.446 | 0.187 | 0.168 | 0.343 | 0.236 | 0.128 | 0.283 | 0.045 | 0.086 | 0.080 | 0.090 | 0.183 | 0.182 | 0.127 | 0.137 |
| HYPER(3KG+2HG) | **0.216** | **0.455** | **0.213** | 0.173 | **0.363** | **0.250** | **0.140** | **0.299** | 0.132 | **0.152** | **0.192** | **0.222** | **0.223** | 0.200 | **0.154** | **0.182** |
| **Finetuned Inference** | | | | | | | | | | | | | | | | |
| ULTRA‡(3KG+2HG) | 0.214 | 0.438 | 0.193 | 0.174 | 0.351 | 0.244 | 0.136 | 0.291 | 0.051 | 0.092 | 0.086 | 0.097 | 0.191 | 0.189 | 0.134 | 0.145 |
| HYPER(3KG+2HG) | 0.217 | 0.456 | 0.209 | 0.176 | 0.347 | 0.243 | 0.158 | 0.275 | 0.169 | 0.171 | 0.194 | 0.210 | 0.225 | 0.234 | 0.166 | 0.210 |

hypergraphs, which are representative of the performance methods relying on end-to-end training. These models, by design, cannot generalize to unseen relations since they explicitly store the trained relation embeddings, and thus have to assign a randomly initialized vector for the representation of the unseen relations. For a fair comparison, we evaluate HYPER(end2end), HYPER models trained directly on the corresponding train set for each dataset.

To also evaluate the pretraining paradigm for foundation models, we include ULTRA‡(3KG/4KG/50KG) from Galkin et al. (2024) as baseline KGFM models[2]. They are pretrained on increasingly large KG corpora and evaluated on reified hypergraphs for tail-only link prediction, following Section 3. To assess the benefits of pretraining on different relational structures, we experimented with four HYPER variants: HYPER(3KG/4KG/50KG), trained only on 3, 4, and 50 *knowledge graph* datasets with the same pre-training mix as ULTRA‡(3KG/4KG/50KG) from Galkin et al. (2024), and HYPER(4HG), trained on four *knowledge hypergraph* datasets (JF17K (Wen et al., 2016), Wikipeople (Guan et al., 2021), FB-AUTO (Fatemi et al., 2020), and M-FB15K (Fatemi et al., 2020)). We further include HYPER(3KG + 2HG), a HYPER model trained on a comprehensive mixture of three knowledge graph (FB15k-237, WN18RR, Codex Medium) and two knowledge hypergraph datasets (JF17K, Wikipeople), aiming to combine the advantages of both types of data, and fine-tuned this checkpoint over the training sets for each downstream task. For a fair comparison, we additionally pretrain ULTRA over the same pretraining mixture on reified hypergraphs and include ULTRA‡(4HG) and ULTRA‡(3KG+2HG).

**Evaluations.** We adopt *filtered ranking protocol*: for each query $q(u_1, \cdots, u_k)$ where $k = \mathtt{ar}(q)$ and for each position $t \leq k$, we replace the $t$-th position by all other entities such that the resulting hyperedges does not appear in training, validation, or testing knowledge hypergraphs. We report Mean Reciprocal Rank (MRR) and provide averaged results for *three* runs for the experiments. We report the standard deviation along with the full tables in Tables 19 and 20 and Table 21. The codebase is provided in `https://github.com/HxyScotthuang/HYPER`. See computation resources used in Appendix D and further experimental details in Appendix G.

## 5.2 NODE-RELATION INDUCTIVE LINK PREDICTION OVER KNOWLEDGE HYPERGRAPHS

**Dataset construction and task settings.** To evaluate the transferability and generalization capabilities of HYPER, we follow the methodology proposed in InGram (Lee et al., 2023) to construct new datasets with varying proportions of unseen relations. We derive these datasets from three

---

[2]We also include additional baseline results for other reification method KGFM in Appendices G.2 and G.3.

hypergraph datasets: JF17K (Wen et al., 2016) (JF), Wikipeople (Guan et al., 2021) (WP), and M-FB15K (Fatemi et al., 2020) (MFB). We also include WD50K (WD) (Galkin et al., 2020), originally a hyper-relational KG, which we convert into a knowledge hypergraph by hashing the main relation and predicates in canonical order. For each source dataset, we create four variants with different percentages of test tuples containing previously unseen relations: 25%, 50%, 75%, and 100%. For instance, JF-25 includes 25% test tuples with unseen relations, while JF-100 contains only entirely unseen relations. This setup[3] allows us to systematically evaluate how models perform under increasingly challenging inductive scenarios. We present all the details in Appendix A.

**Overall performances of HYPER (Q1).** We report model performance across each dataset in Table 2. Note that HYPER and its variants drastically outperform HCNet in node and relation-inductive settings. HCNet relies on learnable embeddings for each relation type and struggles with unseen relations, leading to sharp performance drops under inductive settings. In contrast, HYPER leverages a pretrained relation encoder, enabling strong generalization and robust performance even with entirely unseen relations. Note that fine-tuning HYPER(3KG+2HG) further boosts results, often matching or surpassing HYPER trained from end-to-end. This demonstrates the strong transferability of HYPER's representations and shows that lightweight finetuning on small task-specific datasets can approach end-to-end performance without full retraining.

**Impact on the ratio of known relations (Q2).** We experiment with multiple relation-split settings that vary the proportion of test triplets involving unseen relations, ranging from 25% to 100%. While node-inductive baselines such HCNet already perform poorly under low relational shift (e.g., 25%), their performance degrades substantially as the proportion of unseen relations increases (e.g., 100%), reflecting the difficulty of generalizing to novel relation types. In contrast, HYPER maintains consistently strong performance across all splits,

Table 3: MRR results on node-inductive datasets. ‡ means the model is applied after reification.

| Method | JF-IND | WP-IND | MFB-IND |
|---|---|---|---|
| **End-to-End Inference** | | | |
| HGNN | 0.102 | 0.072 | 0.121 |
| HyperGCN | 0.099 | 0.075 | 0.118 |
| G-MPNN | 0.219 | 0.177 | 0.124 |
| RD-MPNN | 0.402 | 0.304 | 0.122 |
| HCNet | **0.435** | 0.414 | 0.368 |
| HYPER(end2end) | 0.422 | **0.435** | **0.427** |
| **Zero-shot Inference** | | | |
| ULTRA‡(3KG) | 0.321 | 0.305 | 0.277 |
| HYPER(3KG) | 0.263 | 0.259 | 0.184 |
| ULTRA‡(4KG) | 0.065 | 0.123 | 0.096 |
| HYPER(4KG) | 0.266 | 0.231 | 0.120 |
| ULTRA‡(50KG) | 0.007 | 0.029 | 0.026 |
| HYPER(50KG) | 0.302 | 0.253 | 0.248 |
| ULTRA‡(4HG) | 0.397 | 0.319 | 0.264 |
| HYPER(4HG) | 0.403 | 0.375 | **0.497** |
| ULTRA‡(3KG+2HG) | 0.410 | 0.341 | 0.294 |
| HYPER(3KG + 2HG) | **0.459** | **0.415** | 0.404 |
| **Finetuned Inference** | | | |
| ULTRA‡(3KG+2HG) | 0.421 | 0.349 | 0.303 |
| HYPER(3KG + 2HG) | 0.463 | 0.446 | 0.455 |

demonstrating its robustness and ability to generalize effectively under an increased proportion of unseen relations.

**HYPER vs. ULTRA on reified knowledge hypergraph (Q3).** Across all datasets, HYPER consistently outperforms KGFMs like ULTRA‡ on reified hypergraphs. While KGFMs can in principle generalize to binary relations, reified hypergraphs form atypical structures, e.g., tripartite graphs with auxiliary edge nodes, which is not commonly seen in pretraining corpora. Notably, ULTRA‡(50KG), trained on 50 knowledge graphs, performs much worse than the version trained on just 3, and remains substantially behind HYPER(3KG + 2HG). This suggests that increasing the number of training graphs does not close the gap introduced by the lack of explicit hypergraph modeling. Moreover, ULTRA‡(4HG) and ULTRA‡(3KG+2HG) also underperform compared with HYPER(4HG) and HYPER(3KG + 2HG), which are trained on the same pretraining mix containing hypergraph datasets, respectively. While reification makes knowledge hypergraphs compatible with KGFMs, it can hinder generalization: auxiliary edge nodes increase hop distances, inverse relations are modeled ineffectively, and the resulting structures deviate from standard KG pre-training distributions. Together, this leads to inefficient reasoning and weaker performance.

---

[3]These percentages are meaningful only in the end-to-end evaluation setting. In the zero-shot setting, all relation types are unobserved.

Table 4: Averaged zero-shot performance of HYPER(3KG + 2HG) with different positional interaction encoders.

| Model | Total Avg (19 hypergraphs) | |
|---|---|---|
| | MRR | Hits@3 |
| All-one | 0.236 | 0.262 |
| Random | 0.213 | 0.239 |
| Magnitude | 0.227 | 0.251 |
| Sinusoidal | **0.285** | **0.281** |

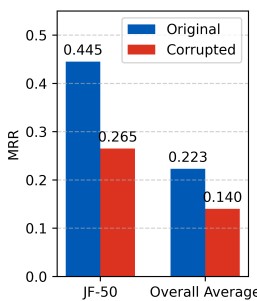

Figure 6: Zero-shot performance of HYPER(3KG + 2HG) over original and corrupted datasets.

**Impact of different pretraining datasets (Q4).** The composition of pretraining data has a noticeable impact on generalization. While HYPER(4HG), pretrained on hypergraph datasets, performs strongly on JF and MFB, both of which contain a large proportion of higher-arity relations, it struggles on WP, which primarily consists of binary edges. Conversely, WP benefits more from pretraining on binary relational graphs, as seen with HYPER(3KG). The best overall performance comes from HYPER(3KG + 2HG), which combines both binary and hypergraph pretraining sources. This suggests that pretraining on diverse relation structures and thus the underlying distribution improves generalization across tasks with varying arities.

## 5.3 NODE INDUCTIVE LINK PREDICTION OVER KNOWLEDGE HYPERGRAPHS

**Settings.** To further assess the applicability of node-inductive link prediction tasks with knowledge hypergraphs, we follow Yadati (2020) and Huang et al. (2025b), and experiment on three existing datasets: JF-IND, WP-IND, and MFB-IND. We compare our models with several existing approaches for inductive link prediction on knowledge hypergraphs. These include HGNN (Feng et al., 2018) and HyperGCN (Yadati et al., 2019), which were originally designed for simple hypergraphs and adapted to knowledge hypergraphs by ignoring relations (Yadati, 2020).

We also compare with G-MPNN (Yadati, 2020) and RD-MPNN (Zhou et al., 2023b), which were modified for inductive settings by replacing learned entity embeddings with a uniform vector, and HCNet (Huang et al., 2025b). We also include the zero-shot performance of standard KGFM ULTRA on the reification of hypergraphs.

**Results and discussion.** Table 3 presents the performance of all models across the node-inductive datasets. We continue to observe that HYPER significantly outperforms prior node-inductive baselines such as HCNet, G-MPNN, and RD-MPNN in the zero-shot setting. Among HYPER variants, even without fine-tuning, pretrained HYPER models achieve strong results. Fine-tuned HYPER further improves performance, achieving the best MRR on JF-IND and WP-IND, and competitive results on MFB-IND compared with HYPER trained end-to-end. Notably, HYPER consistently outperforms ULTRA, which struggles to generalize to the distinct structure of reified hypergraphs. These results confirm HYPER's robust generalization across a variety of datasets.

## 5.4 IMPACT OF POSITIONAL INTERACTION ENCODERS

To evaluate the importance of design choices in the positional interaction encoder $\mathsf{Enc}_{\mathsf{PI}}$ **(Q5)**, we compare HYPER to three alternatives $\mathsf{Enc}_{\mathsf{PI}}$ equipped with different positional encoding schemes: (i) all-one encoding ($p_a = \mathbf{1}^d$), which collapses all positions and violates injectivity; (ii) random encoding ($p_a \sim \mathcal{N}(\mathbf{0}, \mathbf{I}_d)$), which lacks structure and hinders generalization; and (iii) magnitude encoding ($p_a = a\mathbf{1}^d$), which is unbounded and thus unsuitable for MLPs. In contrast, HYPER uses sinusoidal encoding, which is both injective and bounded, enabling effective extrapolation and robust zero-shot performance. As shown in Table 4, sinusoidal encoding yields the best overall performance across 19 hypergraphs, significantly outperforming other schemes in both MRR and Hits@3. This highlights the critical property of injectivity and extrapolation of $\mathsf{Enc}_{\mathsf{PI}}$ in achieving robust zero-shot generalization.

## 5.5 Corruption over Argument Position

To validate the significance of ordered information in knowledge hypergraphs (**Q5**), we conduct an ablation study to corrupt positional information. Specifically, for each of 16 newly proposed datasets, we take the most frequent relation type in each test graph and **randomly** and **inconsistently** permuted the argument positions for 50% of its hyperedges, making the semantic role of each argument position ambiguous. For instance, a hyperedge $r(a, b, c)$ might become $r(b, a, c)$. We evaluate our HYPER(3KG + 2HG) model in the zero-shot setting on these corrupted datasets. Empirically, we observe that when we permute those relations that explicitly stored ordered information, such as cvg.musical_game_song_relationship in JF-50, the performance drops dramatically, as shown in Figure 6. This is because each argument position carries a distinct semantic role (e.g., musical, game, song), and HYPER relies on implicitly learning these roles to generalize. Corrupting this positional structure prevents HYPER from inferring roles for unseen relations, leading to a dramatic decline in performance.

## 6 Conclusion

In this work, we introduced HYPER, the first foundation model for inductive link prediction over knowledge hypergraphs with arbitrary arity, capable of generalizing to both unseen entities and unseen relations. Through extensive experiments on 16 newly constructed and 3 existing inductive benchmarks, we demonstrate that HYPER consistently outperforms state-of-the-art knowledge hypergraph baselines and KGFMs applied to reified hypergraphs, demonstrating its strong generalization across varied domains and relational structures. One limitation of HYPER lies in its computational complexity of relation arity: the number of positional interactions grows quadratically with the arity of each hyperedge. Future work may explore scalable approximations to mitigate the cost. Additionally, KGFMs, such as ULTRA, generally performs better on standard knowledge graph tasks (See Appendix E). Future work may explore bridging the performance gap between higher-order and binary settings.

## Acknowledgment

Bronstein is supported by EPSRC Turing AI World-Leading Research Fellowship No. EP/X040062/1 and EPSRC AI Hub on Mathematical Foundations of Intelligence: An "Erlangen Programme" for AI No. EP/Y028872/1.

## Ethics statement

This work proposes a foundation model for inductive reasoning over knowledge hypergraphs, which may benefit applications in scientific discovery, query answering, and recommendation systems by improving generalization across relational contexts. However, the same capabilities could also be misused for generating or reinforcing biased or spurious inferences when applied to real-world knowledge bases that contain noise, imbalance, or socially sensitive information. Future applications should therefore include safeguards for interpretability and error auditing, especially in domains with fairness or safety considerations. We acknowledge and adhere to the ICLR Code of Ethics.

## Reproducibility Statement

We ensure the reproducibility of our results. A complete description of dataset construction procedures, including inductive splits with varying proportions of unseen relations, is provided in Appendix A. Details of the model architecture, training objectives, and optimization are presented in Section 4 and Appendix G. We further describe efficient implementation details and computational resources in Appendix D. To facilitate verification, we release an anonymous codebase with all scripts for data preprocessing, model training, and evaluation at `https://github.com/HxyScotthuang/HYPER`. A detailed method of relation graph constructions is included in Appendix B. Together, these materials provide all the necessary information to reproduce the experiments and results reported in this paper.

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

# A   DATASET GENERATION DETAILS

## A.1   GENERATING DATASETS FOR NODE AND RELATION-INDUCTIVE LINK PREDICTION

To evaluate our models in an inductive setting, we created multiple dataset variants with different proportions of unseen relations. Our dataset generation process, following InGram (Lee et al., 2023), is detailed in Algorithm 1. This process creates training and inference hypergraphs with controlled percentages of unseen relations in the test set.

---

**Algorithm 1** Generating Datasets for Node and Relation-inductive Link Prediction

---

**Require:** Source knowledge hypergraph $G = (V, E, R)$, number of training entities $n_{\text{train}}$, number of inference entities $n_{\text{test}}$, relation percentage $p_{\text{rel}}$, tuple percentage $p_{\text{tri}}$, seed value
**Ensure:** Training knowledge hypergraph $G_{\text{train}} = (V_{\text{train}}, E_{\text{train}}, R_{\text{train}})$ and Inference knowledge hypergraph $G_{\text{inf}} = (V_{\text{inf}}, E_{\text{inf}}, R_{\text{inf}})$
 1: $G \leftarrow$ Giant connected component of $G$
 2: Randomly split $R$ into $R_{\text{train}}$ and $R_{\text{inf}}$ such that $|R_{\text{train}}| : |R_{\text{inf}}| = (1 - p_{\text{rel}}) : p_{\text{rel}}$
 3: Uniformly sample $n_{\text{train}}$ entities from $V$ and form $V_{\text{train}}$ by taking the sampled entities and their neighbors
 4: $E_{\text{train}} := \{r(v_1, v_2, \ldots, v_n) | v_i \in V_{\text{train}}, r \in R_{\text{train}}, r(v_1, v_2, \ldots, v_n) \in E\}$
 5: $E_{\text{train}} \leftarrow$ Hyperedges in the giant connected component of $E_{\text{train}}$
 6: $V_{\text{train}} \leftarrow$ Entities involved in $E_{\text{train}}$
 7: $R_{\text{train}} \leftarrow$ Relations involved in $E_{\text{train}}$
 8: Let $G'$ be the subgraph of $G$ where the entities in $V_{\text{train}}$ are removed
 9: In $G'$, uniformly sample $n_{\text{test}}$ entities and form $V_{\text{inf}}$ by taking the sampled entities and their neighbors
10: $E_{\text{inf}} := X \cup Y$ such that $|X| : |Y| = (1 - p_{\text{tri}}) : p_{\text{tri}}$ where $X := \{r(v_1, v_2, \ldots, v_n) | v_i \in V_{\text{inf}}, r \in R_{\text{train}}, r(v_1, v_2, \ldots, v_n) \in E\}$ and $Y := \{r(v_1, v_2, \ldots, v_n) | v_i \in V_{\text{inf}}, r \in R_{\text{inf}}, r(v_1, v_2, \ldots, v_n) \in E\}$
11: $E_{\text{inf}} \leftarrow$ Hyperedges in the giant connected component of $E_{\text{inf}}$
12: $V_{\text{inf}} \leftarrow$ Entities involved in $E_{\text{inf}}$
13: $R_{\text{inf}} \leftarrow$ Relations involved in $E_{\text{inf}}$
14: Split $E_{\text{inf}}$ into auxiliary, validation, and test sets with a ratio of 3:1:1

---

The parameter $p_{\text{tri}}$ controls the percentage of test tuples containing unseen relations. For example, when $p_{\text{tri}} = 0.25$, approximately 25% of the tuples in the inference hypergraph contain relations not seen during training. This allows us to systematically evaluate how models perform under increasingly challenging inductive scenarios.

After generating the inference hypergraph, we split it into three disjoint sets: auxiliary (for training), validation, and test sets with a ratio of 3:1:1. For a fair comparison, these sets are fixed and provided to all models.

## A.2   DATASET STATISTICS

Table 5 and Table 6 summarize the statistics of our constructed datasets and the hyperparameters used to generate them, respectively. Additionally, Table 7 presents the arity distribution across these datasets. Together, these tables illustrate that our benchmarks vary significantly in terms of arity, density, and number of relation types, ensuring a diverse and comprehensive evaluation setting.

# B   SPARSE MATRIX MULTIPLICATION FOR COMPUTING POSITIONAL INTERACTION

In this section, we describe the procedure to generalize sparse matrix multiplication to efficiently construct knowledge hypergraphs from hyperedges of arbitrary arity. Unlike knowledge graphs (Galkin et al., 2024), where only two positions (head and tail) exist per relation, resulting in only 4 fundamental relations (head-to-head, head-to-tail, tail-to-head, tail-to-tail), knowledge hy-

Table 5: Statistics of datasets for inductive hypergraph completion. Max arity is shown for training graph and inference graph, respectively.

| Dataset | Train | | | Inference | | | Test | | | Max Arity |
|---------|-------|-------|-------|-------|-------|-------|-------|-------|-------|-----------|
| | $|V|$ | $|R|$ | $|E|$ | $|V|$ | $|R|$ | $|E|$ | $|V|$ | $|R|$ | $|E|$ | |
| JF-25 | 2,616 | 41 | 3,371 | 1,159 | 36 | 1,056 | 209 | 15 | 103 | 5/4 |
| JF-50 | 2,859 | 53 | 3,524 | 1,102 | 37 | 1,292 | 157 | 5 | 109 | 5 |
| JF-75 | 3,129 | 67 | 4,287 | 1,488 | 38 | 1,697 | 225 | 11 | 131 | 5 |
| JF-100 | 2,123 | 48 | 2,449 | 1,696 | 35 | 2,159 | 52 | 5 | 25 | 5 |
| WP-25 | 6,378 | 128 | 7,453 | 2,784 | 66 | 4,794 | 830 | 19 | 959 | 6/4 |
| WP-50 | 7,586 | 155 | 9,536 | 3,608 | 87 | 4,390 | 531 | 29 | 413 | 7/6 |
| WP-75 | 7,787 | 118 | 9,271 | 4,737 | 101 | 6,221 | 629 | 27 | 459 | 6 |
| WP-100 | 7,787 | 118 | 9,271 | 4,891 | 63 | 7,516 | 275 | 15 | 155 | 6 |
| WD-25 | 4,533 | 239 | 5,482 | 3,008 | 191 | 3,106 | 250 | 37 | 148 | 22/5 |
| WD-50 | 3,796 | 162 | 4,147 | 2,303 | 188 | 2,353 | 145 | 30 | 91 | 19/6 |
| WD-75 | 6,518 | 243 | 6,305 | 5,194 | 244 | 5,831 | 547 | 57 | 385 | 22/5 |
| WD-100 | 6,798 | 237 | 7,271 | 3,576 | 105 | 3,951 | 385 | 29 | 282 | 19/4 |
| MFB-25 | 1,266 | 11 | 8,182 | 1,929 | 12 | 2,802 | 146 | 7 | 87 | 3/5 |
| MFB-50 | 1,415 | 11 | 8,409 | 1,528 | 13 | 2,426 | 472 | 10 | 486 | 3/5 |
| MFB-75 | 2,225 | 15 | 5,271 | 1,363 | 16 | 4,008 | 675 | 11 | 803 | 3/4 |
| MFB-100 | 2,013 | 19 | 11,658 | 2,406 | 5 | 4,514 | 808 | 5 | 904 | 3/5 |

Table 6: Hyperparameters used to create fully inductive knowledge hypergraph datasets.

| HP | JF-25 | JF-50 | JF-75 | JF-100 | WP-25 | WP-50 | WP-75 | WP-100 |
|----|-------|-------|-------|--------|-------|-------|-------|--------|
| $n_{\text{train}}$ | 1000 | 1000 | 1200 | 1200 | 900 | 800 | 1000 | 1000 |
| $n_{\text{test}}$ | 900 | 800 | 1200 | 1200 | 800 | 1000 | 1000 | 1000 |
| $p_{\text{rel}}$ | 0.4 | 0.5 | 0.4 | 0.5 | 0.4 | 0.3 | 0.5 | 0.5 |
| $p_{\text{tri}}$ | 0.25 | 0.5 | 0.75 | 1.0 | 0.25 | 0.5 | 0.75 | 1.0 |

| HP | WD-25 | WD-50 | WD-75 | WD-100 | MFB-25 | MFB-50 | MFB-75 | MFB-100 |
|----|-------|-------|-------|--------|--------|--------|--------|---------|
| $n_{\text{train}}$ | 700 | 1000 | 10000 | 10000 | 100 | 100 | 80 | 120 |
| $n_{\text{test}}$ | 1200 | 1000 | 8000 | 8000 | 95 | 85 | 80 | 100 |
| $p_{\text{rel}}$ | 0.25 | 0.5 | 0.5 | 0.5 | 0.5 | 0.6 | 0.5 | 0.5 |
| $p_{\text{tri}}$ | 0.25 | 0.5 | 0.75 | 1.0 | 0.25 | 0.5 | 0.75 | 1.0 |

pergraphs involve $k$ positions per hyperedges, leading to $k^2$ types of possible positional interactions in total.

Given a knowledge hypergraph $G = (V, E, R)$ with $n = |V|$ nodes, $m = |R|$ relations, and maximum arity $k$, we start by representing the knowledge hypergraph via sparse tensors: the edge index $\boldsymbol{E} \in \mathbb{N}^{k \times |E|}$ and corresponding edge types $\boldsymbol{r} \in \mathbb{N}^{|E|}$. Each column of $\boldsymbol{E}$ lists the $k$ participating nodes for a hyperedge, with each edge associated with its relation type.

To encode positional interactions between relations, we perform sparse matrix multiplication in the following steps:

1. For each position $a \in \{1, \cdots, k\}$, we construct sparse matrices $\boldsymbol{E}_a \in \mathbb{R}^{n \times m}$ where each nonzero entry indicates the presence of an entity at position $a$ for a given relation type.

2. For each pair of positions $(a, b) \in \{1, \cdots, k\} \times \{1, \cdots, k\}$, we compute a sparse matrix multiplication:

$$\boldsymbol{A}_{a2b} = \text{spmm}(\boldsymbol{E}_a^\top, \boldsymbol{E}_b) \in \mathbb{R}^{m \times m}.$$

Here, $(\boldsymbol{A}_{a2b})_{i,j}$ is nonzero if there exists an entity that simultaneously plays position $a$ in a hyperedge of relation $i$ and position $b$ in a hyperedge of relation $j$.

This operation systematically captures all intersections between hyperedges that share at least one common node, generalized across different positions.

# C PROOF

We first define isomorphisms and invariants of knowledge hypergraphs, following Huang et al. (2025b;a). The detailed architecture used in the experiments is shown in Appendix G.4.

## C.1 ISOMORPHISMS AND LINK INVARIANTS OF KNOWLEDGE HYPERGRAPHS

**Isomorphisms.** An *(node-relation) isomorphism* from a knowledge hypergraph $G = (V, E, R)$ to a knowledge hypergraph $G' = (V', E', R')$ is a pair of bijections $(\pi, \phi)$, where $\pi : V \to V'$ and $\phi : R \to R'$, such that every fact $r(u_1, \cdots, u_k)$ is in $G$ if and only if the fact $\phi(r)(\pi(u_1), \cdots, \pi(u_k))$ is in $G'$. Two graphs are *isomorphic* if there is an isomorphism between them.

**Invariants.** For $k \geq 1$, a *$k$-ary relation invariant* is a function $\xi$ associating to each knowledge hypergraph $G = (V, E, R)$ a function $\xi(G)$ with domain $R^k$ such that for every pair of isomorphic knowledge hypergraphs $G$ and $G'$, every isomorphism $(\pi, \phi)$, and every tuple $\bar{r} \in R^k$ of relations in $G$, we have

$$\xi(G)(\bar{r}) = \xi(G')(\phi(\bar{r})).$$

A *(link) invariant* is a function $\omega$ assigning to each knowledge hypergraph $G = (V, E, R)$ a function $\omega(G)$ with domain $R \times V^k$ such that for every pair of isomorphic knowledge hypergraphs $G$ and $G'$, every isomorphism $(\pi, \phi)$, and every hyperedge $r(u_1, \ldots, u_k)$ in $G$, we have

$$\omega(G)\big(r(u_1, \ldots, u_k)\big) = \omega(G')\big(\phi(r)(\pi(u_1), \ldots, \pi(u_k))\big).$$

For a query $q = (q, \tilde{\mathbf{u}}, t)$ in $G$, where $q \in R$ is the relation type, $\tilde{\mathbf{u}} = (u_1, \ldots, u_{t-1}, u_{t+1}, \ldots, u_k)$ are the observed arguments, and $t$ is the masked position, we define its image under $(\pi, \phi)$ as

$$(\pi, \phi) \cdot \boldsymbol{q} = \big(\phi(q), (\pi(u_1), \ldots, \pi(u_{t-1}), \pi(u_{t+1}), \ldots, \pi(u_k)), t\big).$$

That is, the relation symbol is mapped via $\phi$ and each entity argument via $\pi$, while the masked position $t$ (and thus the argument order) is preserved.

## C.2 EQUIVARIANCE

We show that HYPER indeed computes a link invariant on knowledge hypergraphs.

**Proposition C.1** (invariance). *Let $G = (V, E, R)$ and $G' = (V', E', R')$ be knowledge hypergraphs, and let $(\pi, \phi)$ be a (node–relation) isomorphism from $G$ to $G'$. Then, the* HYPER *architecture with $T$ layer relation encoders and $L$ layer entity encoders computes link invariant on knowledge hypergraphs, i.e., for every query $q = (q, \tilde{\mathbf{u}}, t)$ in $G$ and every candidate $u_i \in V$,*

$$\text{HYPER}(G)\big(q(u_1, \ldots, u_k)\big) = \text{HYPER}(G')\big(\phi(q)(\pi(u_1), \ldots, \pi(u_k))\big).$$

*Proof.* We show that each stage of HYPER is invariant under the action $(\pi, \phi)$.

**Relation graph construction.** By definition, $G_{\text{rel}} = (V_{\text{rel}}, E_{\text{rel}}, R_{\text{rel}})$ has $V_{\text{rel}} = R$ and contains a directed edge $(r_1, r_2)$ with edge label $(i, j)$ whenever there exist hyperedges $e_1, e_2 \in E$ with $\rho(e_1) = r_1$, $\rho(e_2) = r_2$ and a node $x \in V$ such that $x$ occurs at position $i$ in $e_1$ and at position $j$ in $e_2$. Under an isomorphism map $(\pi, \phi)$, given $G'_{\text{rel}} = (V'_{\text{rel}}, E'_{\text{rel}}, R'_{\text{rel}})$, $\phi$ is a bijection between $R$ and $R'$, and $\pi$ is a bijection between $V$ to $V'$. Thus, the isomorphism map preserves membership and positional indices inside hyperedges. It holds that

$$(r_1, r_2, (i, j)) \in E_{\text{rel}} \iff (\phi(r_1), \phi(r_2), (i, j)) \in E'_{\text{rel}}$$

In particular, the construction is invariant to renaming of relations and respects the same positional labels $(i, j)$. Thus, $G_{\text{rel}}$ and $G'_{\text{rel}}$ are isomorphic via $\phi$ with the same edge labels $(i, j)$.

**Positional-interaction encoder.** By design, $\mathsf{Enc}_{\mathrm{PI}} : \mathbb{N}_{>0} \times \mathbb{N}_{>0} \to \mathbb{R}^d$ maps a position pair $(a,b)$ to $\boldsymbol{x}_{a,b} = \mathrm{MLP}([\boldsymbol{p}_a \| \boldsymbol{p}_b])$, where $\boldsymbol{p}_a, \boldsymbol{p}_b$ are the sinusoidal encodings of $a$ and $b$. Since MLP is shared across all position pairs, $\boldsymbol{x}_{a,b}$ depends only on $(a,b)$. Because edge labels in $G_{\mathrm{rel}}$ and $G'_{\mathrm{rel}}$ are the same pairs $(a,b)$ under $\phi$, we have a label-consistent edge feature assignment: for every edge $(r_1, r_2, (a,b))$ in $G_{\mathrm{rel}}$ and the corresponding $(\phi(r_1), \phi(r_2), (a,b))$ in $G'_{\mathrm{rel}}$, the edge message features coincide, i.e., $\boldsymbol{x}_{a,b} = \boldsymbol{x}'_{a,b}$.

**Relation encoder.** Let $\boldsymbol{h}^{(t)}_{r|q}$ denote the hidden state of the relation node $r \in V_{\mathrm{rel}}$ computed by the HCNet relation encoder at the $t$-th layer, conditioned on query relation $q$. We prove by induction on $t$ that for any isomorphism $(\pi, \phi)$ between $G$ and $G'$,

$$\boldsymbol{h}^{(t)}_{\phi(r)|\phi(q)}(G'_{\mathrm{rel}}) = \boldsymbol{h}^{(t)}_{r|q}(G_{\mathrm{rel}}) \qquad \forall r \in R, \forall t \in \mathbb{N}.$$

Base case: The initialization function $\mathrm{INIT}_{\mathrm{rel}}$ assigns the same parameters to all relations, depending only on whether $r = q$. Because the query relation $q$ maps to $\phi(q)$ under the isomorphism, and all other relations share the same initialization, we have

$$\boldsymbol{h}^{(0)}_{\phi(r)|\phi(q)}(G'_{\mathrm{rel}}) = \boldsymbol{h}^{(0)}_{r|q}(G_{\mathrm{rel}}).$$

Inductive step: Assume the inductive hypothesis holds for layer $t$. We remind that each update of the relation encoder is defined as[4]

$$\boldsymbol{h}^{(t+1)}_{r|q} = \mathrm{UP}_{\mathrm{rel}}\Big(\boldsymbol{h}^{(t)}_{r|q}, \mathrm{AGG}_{\mathrm{rel}}\big(\{\!\!\{\mathrm{MSG}_{\rho(e)}(\{(\boldsymbol{h}^{(t)}_{r'|q}, j) \mid (r',j) \in \mathcal{N}_{\mathrm{rel},i}(e)\}, \boldsymbol{q}) \mid (e,i) \in E_{\mathrm{rel}}(r)\}\!\!\}\big)\Big).$$

Under the isomorphism $\phi$, the neighborhood of each relation node is preserved: for every $(e,i) \in E_{\mathrm{rel}}(r)$ in $G_{\mathrm{rel}}$ there exists a unique $(e',i) \in E'_{\mathrm{rel}}(\phi(r))$ in $G'_{\mathrm{rel}}$, and $\mathcal{N}_{\mathrm{rel},i}(e)$ corresponds bijectively to $\mathcal{N}'_{\mathrm{rel},i}(e')$. Moreover, for each positional label $(a,b)$, the corresponding fundamental relation embedding $\mathbf{x}_{a,b}$ is identical in both graphs since it depends only on $(a,b)$. By the inductive hypothesis, the neighboring states satisfy $\boldsymbol{h}^{(t)}_{\phi(r')|\phi(q)}(G'_{\mathrm{rel}}) = \boldsymbol{h}^{(t)}_{r'|q}(G_{\mathrm{rel}})$ for all $r'$. Because $\mathrm{MSG}_{\rho(e)}$, $\mathrm{AGG}_{\mathrm{rel}}$, and $\mathrm{UP}_{\mathrm{rel}}$ are all shared, differentiable functions applied pointwise across nodes and edges, and $\mathrm{AGG}_{\mathrm{rel}}$ is permutation-invariant, The resulting updates are preserved under the action of $\phi$:

$$\boldsymbol{h}^{(t+1)}_{\phi(r)|\phi(q)}(G'_{\mathrm{rel}}) = \boldsymbol{h}^{(t+1)}_{r|q}(G_{\mathrm{rel}}).$$

**Entity encoder.** This encoder applies an HCNet over the original knowledge hypergraph $G = (V, E, R)$, where each node $v \in V$ represents an entity and each hyperedge $e = r(u_1, \ldots, u_k) \in E$ connects entities according to their argument positions in relation $\rho(e) = r$.
We prove by induction on $\ell$ that for any isomorphism $(\pi, \phi)$ between $G$ and $G'$,

$$\boldsymbol{h}^{(\ell)}_{\pi(v)|(\pi,\phi)\cdot\boldsymbol{q}}(G') = \boldsymbol{h}^{(\ell)}_{v|\boldsymbol{q}}(G) \qquad \forall v \in V, \ \forall \ell \in \mathbb{N}.$$

Base case: The initialization $\mathrm{INIT}$ is shared and depends only on whether the entity participates in the query $\boldsymbol{q}$ and at which position. Since both participation and positional roles are preserved under $(\pi, \phi)$,

$$\boldsymbol{h}^{(0)}_{\pi(v)|(\pi,\phi)\cdot\boldsymbol{q}}(G') = \boldsymbol{h}^{(0)}_{v|\boldsymbol{q}}(G).$$

Inductive step: Assume the inductive hypothesis holds for layer $\ell$. Under $(\pi, \phi)$, each incident pair $(e,i) \in E(v)$ in $G$ corresponds bijectively to $(e',i) \in E'(\pi(v))$ in $G'$, where $e' = \phi(\rho(e))(\pi(u_1), \ldots, \pi(u_k))$ preserves both arity and argument order. The neighborhood mapping $\mathcal{N}_i(e)$ between $\mathcal{N}'_i(e')$ is therefore bijective. For every positional index $j$, the sinusoidal encoding $\boldsymbol{p}_j$ is fixed and identical across graphs. The message computation depends on (i) $\boldsymbol{h}^{(T)}_{\rho(e)|\boldsymbol{q}}$, which equals $\boldsymbol{h}^{(T)}_{\phi(\rho(e))|(\pi,\phi)\cdot\boldsymbol{q}}$ by relation-encoder equivariance, (ii) the neighboring states $\boldsymbol{h}^{(\ell)}_{u|\boldsymbol{q}}$, which

---

[4]Note that $\boldsymbol{h}^{(t+1)}_{r|q} = \boldsymbol{h}^{(t+1)}_{r|\boldsymbol{q}}$ since the position and entity information in query $\boldsymbol{q}$ has been dropped in relation encoder, and thus it is enough to write $\phi(q)$ rather than $(\pi, \phi) \cdot \boldsymbol{q}$.

match $\boldsymbol{h}^{(\ell)}_{\pi(u)|(\pi,\phi)\cdot\boldsymbol{q}}$ by the inductive hypothesis, and (iii) the fixed positional encodings $\boldsymbol{p}_j$. Since $\text{MSG}_{\rho(e)}$, AGG, and UP are shared differentiable functions and AGG is permutation-invariant, the updates are preserved under $(\pi,\phi)$:

$$\boldsymbol{h}^{(\ell+1)}_{\pi(v)|(\pi,\phi)\cdot\boldsymbol{q}}(G') = \boldsymbol{h}^{(\ell+1)}_{v|\boldsymbol{q}}(G).$$

**Decoder.** The unary decoder is a shared map $\text{Dec} : \mathbb{R}^{d(L)} \to [0,1]$ applied to $\boldsymbol{h}^{(L)}_{v|\boldsymbol{q}}$ to score the candidate $v$ for the masked position $t$ in $\boldsymbol{q}$. We have $\boldsymbol{h}^{(L)}_{\pi(v)|(\pi,\phi)\cdot\boldsymbol{q}}(G') = \boldsymbol{h}^{(L)}_{v|\boldsymbol{q}}(G)$, hence

$$\text{Dec}\big(\boldsymbol{h}^{(L)}_{\pi(v)|(\pi,\phi)\cdot\boldsymbol{q}}(G')\big) = \text{Dec}\big(\boldsymbol{h}^{(L)}_{v|\boldsymbol{q}}(G)\big).$$

$\square$

### C.3 REQUIREMENTS OF POSITIONAL INTERACTION ENCODERS

In this section, we justify our choice of positional interaction encoder $\text{MLP}([\boldsymbol{p}_a \| \boldsymbol{p}_b])$ by showing that it satisfies the two key properties: extrapolation and injectivity. We further show that the encoder exhibits smooth dependence on positional indices.

In practice, we implement $\text{Enc}_{\text{PI}}$ as a two-layer multilayer perceptron (MLP) applied to the concatenation of sinusoidal encodings of the two input positions. Let $\boldsymbol{p}_a, \boldsymbol{p}_b \in \mathbb{R}^d$ denote the sinusoidal encodings of positions $a, b \in \mathbb{N}$, defined componentwise by

$$(\boldsymbol{p}_a)_{2i} = \sin\Big(\frac{a}{10000^{2i/d}}\Big), \qquad (\boldsymbol{p}_a)_{2i+1} = \cos\Big(\frac{a}{10000^{2i/d}}\Big), \quad \text{for } i = 0, 1, \ldots, \tfrac{d}{2} - 1,$$

and analogously for $\boldsymbol{p}_b$. The concatenated positional pair is $E(a,b) = [\boldsymbol{p}_a\|\boldsymbol{p}_b] \in [-1,1]^{2d}$, and the positional–interaction encoder outputs $\boldsymbol{x}_{a,b} = \text{MLP}(E(a,b))$.

**Theorem C.2** (Properties of the positional–interaction encoder)**.** *Assume $d$ is even and $10^{8/d}$ is irrational (i.e., $d \notin \{2,4,8\}$). Let $\boldsymbol{p}_a, \boldsymbol{p}_b \in \mathbb{R}^d$ be the sinusoidal positional encodings of positions $a, b \in \mathbb{N}_{>0}$ and $E(a,b) = [\boldsymbol{p}_a\|\boldsymbol{p}_b] \in [-1,1]^{2d}$ as above. Then there exists a choice of parameters of $\text{MLP}$ such that the positional interaction encoder $\text{Enc}_{\text{PI}} : \mathbb{N}^2_{>0} \to \mathbb{R}^m$ satisfies:*

1. ***Injectivity.*** *For all distinct $(a,b), (a',b') \in \mathbb{N}^2_{>0}$,*

$$(a,b) \neq (a',b') \implies \text{Enc}_{\text{PI}}(a,b) \neq \text{Enc}_{\text{PI}}(a',b').$$

2. ***Boundedness of the range.*** *There exists a compact set $K \subset \mathbb{R}^m$ such that*

$$\text{Enc}_{\text{PI}}(a,b) \in K \quad \text{for all } (a,b) \in \mathbb{N}^2_{>0}.$$

3. ***Lipschitz continuity, smoothness.*** *There exists a constant $L > 0$ such that for all $(a,b), (a',b') \in \mathbb{R}^2$,*

$$\big\|\text{Enc}_{\text{PI}}(a,b) - \text{Enc}_{\text{PI}}(a',b')\big\| \leq L\big(|a-a'| + |b-b'|\big).$$

*Proof.* **Injectivity.** First we show that map $a \mapsto \boldsymbol{p}_a$ is injective. Let $\omega_i := 10000^{-2i/d}$. Suppose $\boldsymbol{p}_a = \boldsymbol{p}_b$. Then for each index $i$ there exists $k_i \in \mathbb{Z}$ such that

$$\omega_i(a - b) = 2\pi k_i.$$

Taking $i' = i + 1$, applying the same argument and dividing the two equalities gives

$$\frac{\omega_i}{\omega_{i+1}} = \frac{k_i}{k_{i+1}} \in \mathbb{Q}.$$

But by construction $\frac{\omega_i}{\omega_{i+1}} = 10^{-8/d}$, which is irrational whenever $d \notin \{2,4,8\}$. This is a contradiction unless $a = b$. Hence $a = b$, and the map is injective.

Now we show that the concatenation $E(a,b) = [\boldsymbol{p}_a\|\boldsymbol{p}_b] \in [-1,1]^{2d}$ itself is injective. Suppose $E((a,b)) = E((a',b'))$. By definition of concatenation, the first $d$ dimension coordinates give

$\boldsymbol{p}_a = \boldsymbol{p}_{a'}$ and the last $d$ dimension give $\boldsymbol{p}_b = \boldsymbol{p}_{b'}$. By the previous part, $a = a'$ and $b = b'$. Thus $(a, b) = (a', b')$, so $E$ is injective.

Finally, we show that there exists a parameterization of $\mathrm{MLP}(\boldsymbol{x}) = \mathbf{W}_2\,\sigma(\mathbf{W}_1\boldsymbol{x} + \boldsymbol{b}_1) + \boldsymbol{b}_2$ that is injective. Choose $\mathbf{W}_1 \in \mathbb{R}^{n \times 2d}$ and $\boldsymbol{b}_1 \in \mathbb{R}^n$ so that $\mathbf{W}_1\boldsymbol{x} + \boldsymbol{b}_1 > 0$ coordinatewise for all $\boldsymbol{x} \in [-1, 1]^{2d}$. Then $\sigma$ is the identity on the entire domain, and

$$f(\boldsymbol{x}) = \mathbf{W}_2(\mathbf{W}_1\boldsymbol{x} + \boldsymbol{b}_1) + \boldsymbol{b}_2 = (\mathbf{W}_2\mathbf{W}_1)\,\boldsymbol{x} + (\mathbf{W}_2\boldsymbol{b}_1 + \boldsymbol{b}_2)$$

is an affine map. With $n \geq 2d$ we can pick $\mathbf{W}_1$ to have full column rank $2d$, and with $m \geq 2d$ we can pick $\mathbf{W}_2$ so that $\mathbf{W} := \mathbf{W}_2\mathbf{W}_1$ also has rank $2d$. Hence $\boldsymbol{x} \mapsto \mathbf{W}\boldsymbol{x} + \boldsymbol{c}$ is injective on $\mathbb{R}^{2d}$, thus on $[-1, 1]^{2d}$. Therefore, the class contains injective functions.

**Boundedness of the range.** We need to show that the range of $\mathrm{MLP}(E([\boldsymbol{p}_a\|\boldsymbol{p}_b]))$ is compact. Observe that $\mathrm{MLP}(E(\boldsymbol{p}_a\|\boldsymbol{p}_b)) \subseteq MLP([-1, 1]^{2d})$, since sinusoidal encodings are bounded within $[-1, 1]$ and the concatenation given by $E$ only affects the dimensionality of the embedding. MLP is a continuous function over a compact domain $[-1, 1]^{2d}$, and as a result, its range is compact. Consequently, even for unseen arity indices $(a, b)$, the resulting positional representations remain within the same bounded set as those observed during training.

**Lipschitz continuity (smoothness).** Set

$$C_{\mathrm{pos}} := \sqrt{2\sum_{i=0}^{\frac{d}{2}-1} \omega_i^2}, \quad L_{\mathrm{MLP}} := \|\mathbf{W}_2\|_2\,\|\mathbf{W}_1\|_2, \quad L = C_{\mathrm{pos}}L_{\mathrm{MLP}}.$$

For any $x, y \in \mathbb{R}$ and $\omega > 0$, the mean value theorem gives $|\sin(\omega x) - \sin(\omega y)| \leq \omega|x - y|$ and $|\cos(\omega x) - \cos(\omega y)| \leq \omega|x - y|$ (as sin and cos are all 1-Lipschitz). Hence, with $\omega_i = 10000^{-2i/d}$, we have that

$$\|\boldsymbol{p}_a - \boldsymbol{p}_{a'}\|_2^2 = \sum_{i=0}^{\frac{d}{2}-1} \left(\sin(\omega_i a) - \sin(\omega_i a')\right)^2 + \sum_{i=0}^{\frac{d}{2}-1} \left(\cos(\omega_i a) - \cos(\omega_i a')\right)^2$$
$$\leq \left(2\sum_{i=0}^{\frac{d}{2}-1} \omega_i^2\right) |a - a'|^2.$$

Thus $\|\boldsymbol{p}_a - \boldsymbol{p}_{a'}\|_2 \leq C_{\mathrm{pos}}|a - a'|$, and similarly $\|\boldsymbol{p}_b - \boldsymbol{p}_{b'}\|_2 \leq C_{\mathrm{pos}}|b - b'|$. By triangle inequality on the concatenation,

$$\|E(a, b) - E(a', b')\|_2 = \|[\boldsymbol{p}_a - \boldsymbol{p}_{a'}\|\boldsymbol{p}_b - \boldsymbol{p}_{b'}]\|_2$$
$$\leq \|\boldsymbol{p}_a - \boldsymbol{p}_{a'}\|_2 + \|\boldsymbol{p}_b - \boldsymbol{p}_{b'}\|_2$$
$$\leq C_{\mathrm{pos}}\big(|a - a'| + |b - b'|\big).$$

$\mathrm{MLP}(\boldsymbol{x}) = \mathbf{W}_2\,\sigma(\mathbf{W}_1\boldsymbol{x} + \boldsymbol{b}_1) + \boldsymbol{b}_2$ is Lipschitz with constant $\mathrm{Lip}(\mathrm{MLP}) \leq \|\mathbf{W}_2\|_2\,\mathrm{Lip}(\sigma)\,\|\mathbf{W}_1\|_2 = L_{\mathrm{MLP}}$ since ReLU is 1-Lipschitz ($\sigma = \mathrm{ReLU}$, $\mathrm{Lip}(\sigma) = 1$). Therefore, by compositions of Lipschitz function,

$$\|\mathsf{Enc}_{\mathrm{PI}}(a, b) - \mathsf{Enc}_{\mathrm{PI}}(a', b')\| = \|\mathrm{MLP}(E(a, b)) - \mathrm{MLP}(E(a', b'))\|$$
$$\leq L_{\mathrm{MLP}}\,\|E(a, b) - E(a', b')\|_2$$
$$\leq L_{\mathrm{MLP}}\,C_{\mathrm{pos}}\big(|a - a'| + |b - b'|\big).$$

$\square$

# D  COMPUTATIONAL RESOURCES

All the pretraining experiments is carried out on a single NVIDIA H100 80GB, and the rest of the experiments are carried out using a NVIDIA A10 24GB. Pretraining of HYPER over a single H100 with parameter specified in Appendix G takes 4 days, while fine-tuning and end-to-end training typically require less than 3 hours.

HYPER is implemented primarily using PyTorch and PyTorch Geometric (Fey & Lenssen, 2019), with its core hypergraph message passing implemented via a custom-built Triton kernel[5]. This optimization approximately halves the training time and reduces memory consumption by a factor of five on average. Instead of explicitly materializing all hyperedge messages, as is done in PyTorch Geometric, we directly write neighboring features to the corresponding memory locations during aggregation. While the naive materialization approach incurs $O(k|E|)$ memory complexity, where $k$ denotes the maximum arity and $|E|$ the number of hyperedges, our Triton-based approach achieves $O(|V|)$ memory complexity, depending only on the number of nodes, which enables efficient and scalable training of HYPER models.

## E   ADDITIONAL EXPERIMENTS ON KNOWLEDGE GRAPHS

In addition to the knowledge hypergraph inductive settings, we also evaluate our models on inductive knowledge graph link prediction tasks where both nodes and relations can be unseen during training (**Q6**). This setting presents the most challenging scenario as it requires models to generalize to entirely new knowledge domains with both unseen entities and relation types. We also include inductive node-only knowledge graph link prediction to further strengthen our point.

**Datasets.** For inductive on both nodes and relations task, we includes 13 datasets in INGRAM (Lee et al., 2023): FB-25, FB-50, FB-75, FB-100, WK-25, WK-50, WK-75, WK-100, NL-0, NL-25, NL-50, NL-75, NL-100; and 10 datasets in MTDEA (Zhou et al., 2023a): MT1 tax, MT1 health, MT2 org, MT2 sci, MT3 art, MT3 infra, MT4 sci, MT4 health, Metafram, FBNELL. We also include inductive link prediction on nodes only experiments, containing 12 datasets from GraIL (Teru et al., 2020): WN-v1, WN-v2, WN-v3, WN-v4, FB-v1, FB-v2, FB-v3, FB-v4, NL-v1, NL-v2, NL-v3, NL-v4; 4 datasets from INDIGO (Liu et al., 2021): HM 1k, HM 3k, HM 5k, HM Indigo; and 2 datasets from Nodepiece (Galkin et al., 2022): ILPC Small, ILPC Large.

**Baseline.** We included the zero-shot version of all the models and also include an existing knowledge graph foundation model as baseline, ULTRA (Galkin et al., 2024), shown in Table 8, Table 9. Notably, following standard convention, for every triplet $r(u, v)$ in a knowledge graph, we also include its inverse triplet $r^{-1}(v, u)$, where $r^{-1}$ denotes a newly introduced relation symbol representing the inverse of $r$ for ULTRA. However, HYPER does not need this procedure as the entity encoder employs a variant of HCNet (Huang et al., 2025b), which uses bi-directional message-passing and automatically considers the message from the inverse direction.

**Results and Discussion.** We observe that ULTRA generally performs better on standard inductive link prediction on knowledge graphs, although HYPER is still competitive overall. Across both node-only and node-and-relation inductive benchmarks, HYPER performs on par with ULTRA, and often outperforms it on datasets with higher relational diversity or structure. These results demonstrate that the architectural inductive bias of HYPER, originally designed for knowledge hypergraphs, also transfers well to standard knowledge graphs, without compromising generalization ability.

## F   COMPLEXITY AND SCALABILITY ANALYSIS OF HYPER

To answer **Q7**, we first examine the theoretical computational complexity of HYPER in Appendix F.1, then present its empirical scalability results when applying on FB15k-237 Appendix F.2.

### F.1   THEORETICAL COMPUTATIONAL COMPLEXITY

In this section, we analyze the computational complexity of HYPER. Let $G = (V, E, R)$ denote the input knowledge hypergraph, where $n = |V|$, $m = |E|$, and $|R|$ are the number of entities, hyperedges, and relation types, respectively. Let $k$ be the maximum arity of $R$, $d$ the hidden dimension, and $T$ the number of message-passing layers in the relation encoder, and denote $L$ as the number of message-passing layers in the entity encoder.

**Relation Graph Construction**   The complexity of generating the relation graph in HYPER arises from computing pairwise positional interactions between relation types across hyperedges of ar-

---

[5]https://github.com/triton-lang/triton

bitrary arity. Unlike knowledge graphs, where each relation involves exactly two fixed positions (head and tail), knowledge hypergraphs induce up to $k^2$ positional interaction types for a maximum arity $k$. For each position $a \in \{1, \ldots, k\}$, we construct sparse matrices $\boldsymbol{E}_a \in \mathbb{R}^{n \times m}$ that index entities by their position and relation type. Then, for every pair $(a, b)$, we perform a sparse matrix multiplication: $\mathrm{spmm}(\boldsymbol{E}_a^\top, \boldsymbol{E}_b)$. Each such multiplication has a worst-case complexity of $\mathcal{O}(\mathrm{nnz}(\boldsymbol{E}_a^\top) \cdot \mathrm{nnz}(\boldsymbol{E}_b))$, where $\mathrm{nnz}(\cdot)$ denotes the number of nonzero entries. Since there are $k^2$ position pairs, the total time complexity of constructing the relation graph becomes

$$\mathcal{O}(k^2 \cdot \max_{\{a,b\}} \{\mathrm{nnz}(\boldsymbol{E}_a^\top) \cdot \mathrm{nnz}(\boldsymbol{E}_b)\}).$$

In practice, this is significantly accelerated by sparse tensor and batching across position pairs. Without sparse matrix multiplication, the naive construction would require iterating over all hyperedge pairs, resulting in $\mathcal{O}(k^2 |E|^2)$ complexity, which is infeasible for large-scale datasets.

Additionally, for the positional interaction encoders, we associate a positional encoding vector $\mathsf{Enc}_{\mathrm{PI}}((a, b)) \in \mathbb{R}^d$. This construction requires $\mathcal{O}(k^2 d)$ time and space to compute and store.

**Relation Encoder** The relation encoder in HYPER performs $T$ layers of message passing over the relation graph $G_{\mathrm{rel}} = (V_{\mathrm{rel}}, E_{\mathrm{rel}}, R_{\mathrm{rel}})$, as constructed before.

There are at most $k^2$ position pairs per pair of relation types, where $k$ is the maximum arity, so the total number of edges is bounded by

$$|E_{\mathrm{rel}}| = \mathcal{O}(|R|^2 k^2).$$

In each message passing layer, each relation node aggregates messages from up to $|R|^2 k^2$ neighbors, with each edge contributing a message via the corresponding positional interaction embedding $\boldsymbol{x}_{a,b} = \mathsf{Enc}_{\mathrm{PI}}((a, b)) \in \mathbb{R}^d$. Each node then applies an update with cost $\mathcal{O}(d^2)$. Thus, the total complexity of the relation encoder over $T$ layers is

$$\mathcal{O}\left(T(|R|^2 k^2 d + |R| d^2)\right).$$

**Entity Encoder** After obtaining relation embeddings from the relation encoder, HYPER applies $L$ layers of conditional message passing over the original knowledge hypergraph $G = (V, E, R)$ using HCNet (Huang et al., 2025b). In each layer, every entity $v \in V$ aggregates messages from its incident hyperedges $e \in E(v)$, where each hyperedge contributes a query-conditioned message, taking $\mathcal{O}(L(k|E|d)$, that incorporates its relation embedding $h_{\rho(e)|q}^{(T)} \in \mathbb{R}^d$, followed by a relation-specific MLP, which takes $\mathcal{O}(L|R|d^2)$. Each entity then updates its representation through a neural update function with cost $\mathcal{O}(d^2)$.

The total complexity of the entity encoder over $L$ layers is thus

$$\mathcal{O}(L(k|E|d + |V|d^2 + |R|d^2)).$$

## F.2    SCALABILITY ANALYSIS

To empirically assess the scalability of HYPER, we compare HYPER with ULTRA, a prominent knowledge graph foundation model, and HCNet, a state-of-the-art node-inductive method on link prediction with knowledge hypergraph. All experiments are conducted on the *transductive* knowledge graph dataset FB15k-237 using a batch size of 64 to ensure a fair comparison among all three methods. We summarize the model parameter size, training/inference times, and GPU memory usage.

Compared with HCNets, HYPER's training and inference times are approximately doubled since HYPER employs *two* HCNet encoders, one for relations and one for entities. We argue that this overhead represents a reasonable trade-off for the substantial performance improvements and stronger inductive generalization demonstrated by HYPER compared with HCNets.

Compared with ULTRA, the main bottleneck of scalability is the complex modeling of knowledge graphs as knowledge hypergraphs. These differences essentially reduce to the difference between HCNet and NBFNets. For a detailed discussion, we refer the reader to Huang et al. (2025b).

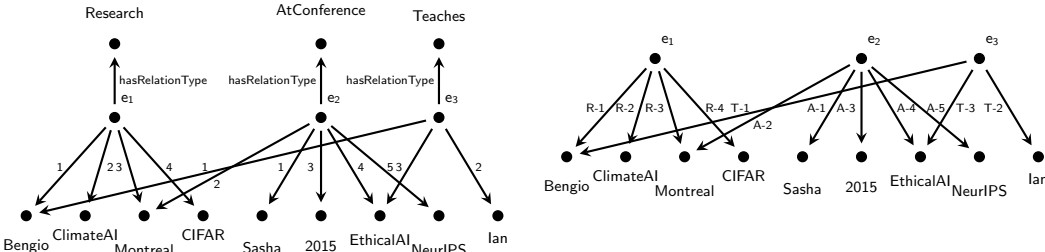

Figure 7: Two different reification KG corresponding to the knowledge hypergraph $G_{\text{train}}$ from Fig 1. On the left figure (†), $i$ abbreviates hasEntity$_i$. On the right figure (‡), R-i abbreviates Research-i, similarly for A as AtConference, and T as Teaches.

## G  FURTHER EXPERIMENTAL DETAILS

In this section, we provide detailed experimental configurations and dataset statistics. In particular, Table 13 summarizes the training corpora used for each model variant across knowledge graph and knowledge hypergraph settings. Tables 14 and 15 present arity distributions and structural statistics for the node-inductive datasets, while Table 16 reports the corresponding statistics for pretraining datasets. For inductive link prediction involving unseen entities and relations, we provide comprehensive dataset breakdowns in Tables 17 and 18.

We also include the complete performance tables together with standard deviation for the node-inductive and node-relation inductive settings shown in Tables 19 to 21, respectively. Table 22 lists all hyperparameter choices used for pretraining, fine-tuning, and end-to-end training of HYPER. Finally, Table 23 specifies the dataset-specific training schedules for each experimental regime.

### G.1  HYPERPARAMETER DETAILS FOR BASELINES

For G-MPNNs, we adopt the best-performing hyperparameters from the original codebase. Specifically, we set the input dimension $d = 64$, hidden dimension $h = 150$, and dropout rate to $0.5$. We use a training batch size $b = 128$, evaluation batch size $B = 4$, and negative sampling ratio $nr = 10$. The learning rate is set to $0.0005$, and models are trained for up to 5000 epochs with validation evaluated every 5 epochs. Aggregation is performed using the max.

For HCNet, we use a 6-layer encoder with an input dimension of 64 and a hidden dimension of 64 for all layers. We adopt sum as the aggregation function and enable shortcut connections to facilitate training. Optimization is performed using AdamW with a learning rate of $5 \times 10^{-4}$. Training is conducted with a batch size of 8, using the same number of epochs and batches per epoch as HYPER, with validation performed every 100 steps. The model is trained with 256 adversarial negatives sampled per positive example, and strict negative sampling is enforced to prevent overlap with true triples.

### G.2  ADDITIONAL BASELINES FOR KGFMS

To provide a more comprehensive evaluation of Knowledge Graph Foundation Models (KGFMs), we expanded our analysis beyond ULTRA, which served as the representative KGFM in the main paper. We additionally evaluated KG-ICL (Cui et al., 2024), a model that has demonstrated strong performance on inductive knowledge graph completion tasks. For KG-ICL, we performed zero-shot experiments across all 16 proposed datasets after reification with 3 variants of KG-ICL using 4 Layer, 5 Layer, and 6 Layer encoders, respectively. We highlight that KG-ICL learns over different pretraining mix than ULTRA (FB-v1, NL-v1, and Codex Small). The results, summarized in Table 12, indicate that KG-ICL's performance is substantially lower than that of HYPER, further supporting the conclusion that existing KGFMs exhibit limited generalization on hypergraph structures.

### G.3 Impact of Reification Methods on KGFMs

We proposed an alternative reification used in Section 3, where each hyperedge becomes an auxiliary node with outgoing edges hasEntity_$i$ to its arguments and one hasRelationType edge to the relation node (left of Fig. 7), noted as (†)). After edge_id node is generated for each edge, we then generate binary edges of the form hasEntity$_i$(edge_id, $u_i$) for each $i \in [k]$ to capture the positions of the entities in the relation. Finally, we add the original relation $r$ as a node to the KG and add an edge hasRelationType(edge_id, $r$).

Intuitively, † adds a two-hop detour (edge_id → hasRelationType → $r$), unlike ‡ which encodes the relation and positional information together at the edge types itself. This generally decreases the performance by over-simplifying the graph structure and disallows KGFM to use different representations for different relations, making the model less efficient. As a result, we opt not to use this in reification in the main experiments.

Table 11 shows two main trends. *(i)* With KG-only pretraining (3KG/4KG/50KG), ULTRA‡ is volatile with the relation explosion and we do not observe scaling behavior, while ULTRA† is generally more robust to the reified knowledge hypergraphs and shows a more consistent trend as number of pretraining mix grows. *(ii)* When pretraining includes hypergraphs (4HG), (3KG + 2HG), exposing role information at the edge type helps a lot: ULTRA‡ typically matches or exceeds ULTRA† across splits.

Nevertheless, all of the reficiation schemes with ULTRA underperform compared with HYPER, which avoids reification entirely by design and remains strongest by operating directly on hypergraph.

### G.4 Architecture Choices of Hyper

Both the relation and entity encoders in HYPER follow the design based on HCNets (Huang et al., 2025b), with a minor variant on the relation-specific message functions.

**Positional Interactions.** In practice, we implement $\mathsf{Enc}_{\mathrm{PI}}$ as a two-layer multilayer perceptron (MLP) over concatenated sinusoidal encodings of the input positions. Let $\boldsymbol{p}_a, \boldsymbol{p}_b \in \mathbb{R}^d$ denote the sinusoidal positional encodings of positions $a$ and $b$, respectively. Specifically, let $\boldsymbol{p}_a, \boldsymbol{p}_b \in \mathbb{R}^d$ denote the sinusoidal positional encodings of positions $a, b \in \mathbb{N}$, defined componentwise as

$$(\boldsymbol{p}_a)_{2i} = \sin\left(\frac{a}{10000^{2i/d}}\right), \qquad (\boldsymbol{p}_a)_{2i+1} = \cos\left(\frac{a}{10000^{2i/d}}\right), \quad \text{for } i = 0, 1, \ldots, \frac{d}{2} - 1,$$

and similarly for $\boldsymbol{p}_b$.

Then, the embedding corresponding to the interaction $(a, b)$ is computed as $\mathbf{x}_{a,b} = \mathrm{MLP}([\boldsymbol{p}_a \,\|\, \boldsymbol{p}_b])$, where MLP denotes a shared two-layer feedforward network with ReLU activations. This produces a dense embedding that captures the interaction between the two positions. Empirically, we find that this instantiation of $\mathsf{Enc}_{\mathrm{PI}}$ enables strong generalization across knowledge hypergraphs with varying arities and relational structures.

**Relation Encoder.** The relation encoder applies an HCNet over the constructed relation graph $(V_{\mathrm{rel}}, E_{\mathrm{rel}}, R_{\mathrm{rel}})$. Here, each node $r \in V_{\mathrm{rel}}$ represents a relation type in $G$, and an edge captures the induced interactions among relations. For each relation $r \in V_{\mathrm{rel}}$, HCNet iteratively updates its representation $\boldsymbol{h}_{r|\boldsymbol{q}}^{(t)}$ as:

$$\boldsymbol{h}_{r|\boldsymbol{q}}^{(0)} = \mathrm{INIT}_{\mathrm{rel}}(r, \boldsymbol{q}),$$

$$\boldsymbol{h}_{r|\boldsymbol{q}}^{(t+1)} = \mathrm{UP}_{\mathrm{rel}}\Big(\boldsymbol{h}_{r|\boldsymbol{q}}^{(t)}, \mathrm{AGG}_{\mathrm{rel}}\big(\{\!\!\{ \mathrm{MSG}_{\rho(e)}(\{(\boldsymbol{h}_{r'|\boldsymbol{q}}^{(t)}, j) \mid (r', j) \in \mathcal{N}_{\mathrm{rel}\,i}(e)\}, \boldsymbol{q}) \mid (e, i) \in E_{\mathrm{rel}}(r)\}\!\!\}\big)\Big),$$

where $E_{\mathrm{rel}}(r)$ is the set of edge-position pairs incident to $r$, and $\mathcal{N}_{\mathrm{rel}\,i}(e)$ is the positional neighborhood of hyperedge $e$ at position $i$. After $T$ layers, we obtain the final relation encoding $\boldsymbol{h}_{r|\boldsymbol{q}}^{(T)}$. Here, $\mathrm{INIT}_{\mathrm{rel}}$, $\mathrm{UP}_{\mathrm{rel}}$, $\mathrm{AGG}_{\mathrm{rel}}$, and $\mathrm{MSG}_{\rho(e)}$ are differentiable initialization, update, aggregation, and fundamental relation-specific message functions, respectively. The initialization function $\mathrm{INIT}_{\mathrm{rel}}$ is designed to satisfy *generalized target node distinguishability* as formalized in Huang et al. (2025b).

Empirically, we initialize the query node $q \in V_{\text{rel}}$ with an all-one vector and all other relation nodes with zero vectors.

In the experiments, we adopt the fundamental relation-specific message function $\text{MSG}_{r_{\text{fund}}}$ using the fundamental relation embedding $\mathbf{r}_{a,b}$. Specifically, given a set of neighbor features $\{(\boldsymbol{h}_{w|\boldsymbol{q}}^{(\ell)}, j) \mid (w, j) \in \mathcal{N}_{\text{rel}i}(e)\}$ for hyperedge $e$ and center position $i$, the message is computed as:

$$\text{MSG}_{r_{a,b}}\left(\{(\boldsymbol{h}_{w|\boldsymbol{q}}^{(t)}, j) \mid (w, j) \in \mathcal{N}_{\text{rel}i}(e)\}\right) = \left(\bigodot_{j \neq i}\left(\alpha^{(t)} \boldsymbol{h}_{e(j)|\boldsymbol{q}}^{(t)} + (1 - \alpha^{(t)})\boldsymbol{p}_j\right)\right) \odot \mathbf{x}_{a,b},$$

where $\odot$ is the elemental-wise multiplication, $\alpha^{(\ell)}$ is a learnable scalar, $\boldsymbol{p}_j$ is the sinusoidal positional encoding at position $j$, and $\mathbf{x}_{a,b}$ is the fundamental relation embedding computed as described earlier.

**Entity Encoder.**   In the context of the entity encoder, we apply a separate HCNet over the original knowledge hypergraph $G = (V, E, R)$. Each node $v \in V$ aggregates information from its incident hyperedges, incorporating the relation embeddings $\boldsymbol{h}_{r|\boldsymbol{q}}^{(T)}$ obtained from the relation encoder.

Given a query $\boldsymbol{q} = (q, \tilde{\boldsymbol{u}}, t)$, where $\tilde{\boldsymbol{u}} = (u_1, \ldots, u_k)$ denotes the entities in the hyperedge and $t$ is the target position, each node $v \in V$ receives an initial representation defined as:

$$\boldsymbol{h}_{v|\boldsymbol{q}}^{(0)} = \sum_{i \neq t} \mathbb{1}_{v = u_i} \cdot (\boldsymbol{p}_i + \boldsymbol{h}_{q|\boldsymbol{q}}^{(T)}),$$

where $\boldsymbol{p}_i \in \mathbb{R}^d$ is the positional encoding at position $i$.

HYPER then iteratively updates the node representations $\boldsymbol{h}_{v|\boldsymbol{q}}^{(\ell)}$ as:

$$\boldsymbol{h}_{v|\boldsymbol{q}}^{(0)} = \text{INIT}(v, \boldsymbol{q}),$$
$$\boldsymbol{h}_{v|\boldsymbol{q}}^{(\ell+1)} = \text{UP}\left(\boldsymbol{h}_{v|\boldsymbol{q}}^{(\ell)}, \text{AGG}\left(\{\!\{\text{MSG}_{\rho(e)}(\{(\boldsymbol{h}_{w|\boldsymbol{q}}^{(\ell)}, j) \mid (w, j) \in \mathcal{N}_i(e)\}, \boldsymbol{h}_{\rho(e)|\boldsymbol{q}}^{(T)}, \boldsymbol{q}) \mid (e, i) \in E(v)\}\!\}\right)\right).$$

where INIT, UP, AGG, and $\text{MSG}_r$ are differentiable initialization, update, aggregation, and relation-specific message functions, respectively, with INIT satisfying *generalized targets node distinguishability* (Huang et al., 2025b). After $L$ layers of message passing, we obtain the final entity encoding $\boldsymbol{h}_{v|\boldsymbol{q}}^{(L)}$. A final unary decoder $\text{Dec} : \mathbb{R}^{d(L)} \to [0, 1]$ predicts the score for completing the missing position $t$ in the query $\boldsymbol{q}$.

Empirically, we select the relation-specific message function $\text{MSG}_{\rho(e)}$ to be

$$\text{MSG}_r\left(\{(\boldsymbol{h}_{w|\boldsymbol{q}}^{(\ell)}, j) \mid (w, j) \in \mathcal{N}_i(e)\}\right) = \left(\bigodot_{j \neq i}\left(\alpha^{(\ell)} \boldsymbol{h}_{e(j)|\boldsymbol{q}}^{(\ell)} + (1 - \alpha^{(\ell)})\boldsymbol{p}_j\right)\right) \odot \text{MLP}^{(\ell)}(\boldsymbol{h}_{\rho(e)|\boldsymbol{q}}^{(T)}),$$

where additionally $\text{MLP}^{(\ell)}$ is a 2-layer MLP with ReLU to transform the relation representation most suitable for each specific layer during message passing.

**Update.**   We use summation as the aggregation operator for both relation and entity nodes. Each node updates its representation via a two-layer MLP applied to the concatenation of its current state and the aggregated message:

$$\boldsymbol{h}_{v|\boldsymbol{q}}^{(\ell+1)} = \text{MLP}^{(\ell)}\left([\boldsymbol{h}_{v|\boldsymbol{q}}^{(\ell)} \| \text{AGGREGATE}_{v|\boldsymbol{q}}^{(\ell)}]\right),$$

where $\text{AGGREGATE}_{v|\boldsymbol{q}}^{(\ell)}$ denotes the sum of incoming messages to node $v$ under query $\boldsymbol{q}$ at layer $\ell$, and $\|$ represents vector concatenation.

**Other.**   We also apply layer normalization and shortcut connections after aggregation and before the ReLU activation in both encoders.

## G.5 TRAINING OBJECTIVE

Following prior work (Huang et al., 2025b), we train HYPER under the *partial completeness assumption* (Galárraga et al., 2013), where each $k$-ary fact $q(u_1, \ldots, u_k)$ is used to generate training samples by randomly masking one position $1 \leq t \leq k$. Given a query $\boldsymbol{q} = (q, \tilde{\boldsymbol{u}}, t)$, we model the conditional probability of entity $v \in V$ filling the missing position as $p(v|\boldsymbol{q}) = \sigma(\text{Dec}(\boldsymbol{h}_{v|\boldsymbol{q}}^{(L)}))$, where Dec is a two-layer MLP and $\sigma$ denotes the sigmoid activation. We optimize the following self-adversarial negative sampling loss (Sun et al., 2019):

$$\mathcal{L}(v|\boldsymbol{q}) = -\log p(v|\boldsymbol{q}) - \sum_{i=1}^{n} w_{i,\alpha} \log(1 - p(v_i'|\boldsymbol{q})),$$

where $v_i'$ are corrupted negative samples, $n$ is the number of negatives per query, $\alpha$ being the adversarial temperature, and $w_{i,\alpha}$ are the importance weights defined by

$$w_{i,\alpha} = \text{Softmax}\left(\frac{\log(1 - p(v_i'|\boldsymbol{q}))}{\alpha}\right).$$

To mitigate overfitting, we exclude edges that directly connect query node pairs during training. The best model checkpoint is selected based on validation performance. Following the implementation of ULTRA (Galkin et al., 2024), for pertaining over multiple knowledge graph and knowledge hypergraphs, for each batch, we sample from one of the pretrained (hyper)graphs with probability proportional to the number of edges it contains.

Table 7: Arity distribution across node-relation inductive datasets.

| Dataset | Arity | Training Graph | Inference Graph | Training % | Inference % |
|---|---|---|---|---|---|
| JF-25 | 2 | 1585 | 266 | 47.02% | 25.19% |
| | 3 | 1441 | 670 | 42.75% | 63.45% |
| | 4 | 326 | 120 | 9.67% | 11.36% |
| | ≥5 | 19 | 0 | 0.56% | 0.00% |
| JF-50 | 2 | 1942 | 321 | 55.11% | 24.85% |
| | 3 | 1297 | 692 | 36.80% | 53.56% |
| | 4 | 285 | 279 | 8.09% | 21.59% |
| | ≥5 | 0 | 0 | 0.00% | 0.00% |
| JF-75 | 2 | 2641 | 824 | 61.60% | 48.56% |
| | 3 | 848 | 846 | 19.78% | 49.85% |
| | 4 | 779 | 27 | 18.17% | 1.59% |
| | ≥5 | 19 | 0 | 0.44% | 0.00% |
| JF-100 | 2 | 1349 | 1637 | 55.08% | 75.82% |
| | 3 | 570 | 283 | 23.27% | 13.11% |
| | 4 | 511 | 159 | 20.87% | 7.36% |
| | ≥5 | 19 | 80 | 0.78% | 3.71% |
| WD-25 | 2 | 4,331 | 2,799 | 79.00% | 90.12% |
| | 3 | 612 | 162 | 11.16% | 5.22% |
| | 4 | 463 | 144 | 8.45% | 4.64% |
| | ≥5 | 76 | 1 | 1.39% | 0.03% |
| WP-50 | 2 | 7709 | 4355 | 80.84% | 99.20% |
| | 3 | 1106 | 28 | 11.60% | 0.64% |
| | 4 | 667 | 3 | 6.99% | 0.07% |
| | ≥5 | 54 | 4 | 0.57% | 0.09% |
| WP-75 | 2 | 6471 | 6121 | 69.80% | 98.39% |
| | 3 | 1725 | 82 | 18.61% | 1.32% |
| | 4 | 1026 | 15 | 11.07% | 0.24% |
| | ≥5 | 49 | 3 | 0.53% | 0.05% |
| WP-100 | 2 | 6471 | 7413 | 69.80% | 98.63% |
| | 3 | 1725 | 91 | 18.61% | 1.21% |
| | 4 | 1026 | 6 | 11.07% | 0.08% |
| | ≥5 | 49 | 6 | 0.53% | 0.08% |
| WD-25 | 2 | 3,680 | 1,941 | 87.60% | 93.81% |
| | 3 | 211 | 114 | 5.02% | 5.51% |
| | 4 | 279 | 12 | 6.64% | 0.58% |
| | ≥5 | 31 | 2 | 0.74% | 0.10% |
| WD-50 | 2 | 3,238 | 2,127 | 78.08% | 90.40% |
| | 3 | 417 | 86 | 10.06% | 3.65% |
| | 4 | 438 | 136 | 10.56% | 5.78% |
| | ≥5 | 54 | 4 | 1.30% | 0.17% |
| WD-75 | 2 | 4,900 | 5,669 | 77.72% | 97.22% |
| | 3 | 769 | 139 | 12.20% | 2.38% |
| | 4 | 548 | 22 | 8.69% | 0.38% |
| | ≥5 | 88 | 1 | 1.40% | 0.02% |
| WD-100 | 2 | 5,858 | 3,631 | 80.57% | 91.90% |
| | 3 | 906 | 186 | 12.46% | 4.71% |
| | 4 | 397 | 134 | 5.46% | 3.39% |
| | ≥5 | 110 | 0 | 1.51% | 0.00% |
| MFB-25 | 2 | 137 | 1555 | 1.67% | 55.50% |
| | 3 | 8045 | 831 | 98.33% | 29.66% |
| | 4 | 0 | 0 | 0.00% | 0.00% |
| | ≥5 | 0 | 416 | 0.00% | 14.85% |
| MFB-50 | 2 | 149 | 1400 | 1.77% | 57.71% |
| | 3 | 8260 | 756 | 98.23% | 31.16% |
| | 4 | 0 | 0 | 0.00% | 0.00% |
| | ≥5 | 0 | 270 | 0.00% | 11.13% |
| MFB-75 | 2 | 2774 | 368 | 52.63% | 9.18% |
| | 3 | 2497 | 3639 | 47.37% | 90.79% |
| | 4 | 0 | 1 | 0.00% | 0.02% |
| | ≥5 | 0 | 0 | 0.00% | 0.00% |
| MFB-100 | 2 | 726 | 3234 | 6.23% | 71.64% |
| | 3 | 10932 | 370 | 93.77% | 8.20% |
| | 4 | 0 | 0 | 0.00% | 0.00% |
| | ≥5 | 0 | 910 | 0.00% | 20.16% |

Table 8: Zero-shot experiment results on node and relation inductive knowledge graph datasets

| Method | FB-25 | | FB-50 | | FB-75 | | FB-100 | | WK-25 | | WK-50 | | WK-75 | | WK-100 | |
|---|---|---|---|---|---|---|---|---|---|---|---|---|---|---|---|---|
| | MRR | H@10 | MRR | H@10 | MRR | H@10 | MRR | H@10 | MRR | H@10 | MRR | H@10 | MRR | H@10 | MRR | H@10 |
| ULTRA(3KG) | **0.388** | **0.640** | **0.338** | **0.543** | **0.403** | **0.604** | **0.449** | **0.642** | **0.316** | **0.532** | **0.166** | **0.324** | **0.365** | **0.537** | 0.164 | **0.286** |
| HYPER(3KG) | 0.372 | 0.614 | 0.313 | 0.513 | 0.373 | 0.568 | 0.412 | 0.598 | 0.276 | 0.410 | 0.145 | 0.281 | 0.334 | 0.460 | **0.171** | 0.271 |
| HYPER(4HG) | 0.277 | 0.538 | 0.225 | 0.427 | 0.287 | 0.503 | 0.336 | 0.567 | 0.215 | 0.422 | 0.117 | 0.245 | 0.280 | 0.491 | 0.125 | 0.247 |
| HYPER(3KG + 2HG) | 0.382 | 0.635 | 0.326 | 0.535 | 0.389 | 0.598 | 0.434 | 0.632 | 0.281 | 0.428 | 0.158 | 0.280 | 0.365 | 0.522 | 0.160 | 0.280 |

| Method | NL-25 | | NL-50 | | NL-75 | | NL-100 | | MT1-tax | | MT1-health | | MT2-org | | MT2-sci | |
|---|---|---|---|---|---|---|---|---|---|---|---|---|---|---|---|---|
| | MRR | H@10 | MRR | H@10 | MRR | H@10 | MRR | H@10 | MRR | H@10 | MRR | H@10 | MRR | H@10 | MRR | H@10 |
| ULTRA(3G) | **0.395** | **0.569** | **0.407** | **0.570** | **0.368** | **0.547** | 0.471 | 0.651 | 0.224 | 0.305 | 0.298 | 0.374 | **0.095** | **0.159** | **0.258** | 0.354 |
| HYPER(3KG) | 0.321 | 0.550 | 0.350 | 0.520 | 0.320 | 0.483 | 0.415 | 0.627 | **0.234** | 0.306 | **0.361** | **0.431** | 0.088 | 0.142 | 0.256 | 0.339 |
| HYPER(4HG) | 0.214 | 0.431 | 0.226 | 0.480 | 0.252 | 0.455 | 0.333 | 0.618 | 0.200 | 0.274 | 0.266 | 0.358 | 0.063 | 0.116 | 0.195 | 0.320 |
| HYPER(3KG + 2HG) | 0.360 | 0.558 | 0.376 | 0.547 | 0.342 | 0.540 | **0.473** | **0.685** | 0.204 | **0.396** | 0.222 | 0.399 | 0.087 | 0.149 | **0.258** | **0.428** |

| Method | MT3-art | | MT3-infra | | MT4-sci | | MT4-health | | Metafam | | FBNELL | | NL-0 | | Average | |
|---|---|---|---|---|---|---|---|---|---|---|---|---|---|---|---|---|
| | MRR | H@10 | MRR | H@10 | MRR | H@10 | MRR | H@10 | MRR | H@10 | MRR | H@10 | MRR | H@10 | MRR | H@10 |
| ULTRA(3KG) | 0.259 | 0.402 | **0.619** | **0.755** | **0.274** | **0.449** | **0.624** | **0.737** | 0.238 | 0.644 | **0.485** | **0.652** | **0.342** | 0.523 | **0.345** | 0.513 |
| HYPER(3KG) | 0.257 | 0.402 | 0.562 | 0.695 | 0.259 | 0.415 | 0.547 | 0.723 | 0.395 | 0.804 | 0.447 | 0.617 | 0.312 | 0.501 | 0.318 | 0.492 |
| HYPER(4HG) | 0.152 | 0.265 | 0.363 | 0.451 | 0.232 | 0.412 | 0.380 | 0.556 | 0.191 | 0.606 | 0.320 | 0.537 | 0.171 | 0.393 | 0.228 | 0.419 |
| HYPER(3KG + 2HG) | **0.270** | **0.425** | 0.573 | 0.716 | 0.270 | 0.441 | 0.560 | 0.724 | **0.457** | **0.875** | 0.450 | 0.639 | 0.334 | **0.526** | 0.336 | **0.520** |

Table 9: Zero-shot experiment results on node inductive knowledge graph datasets. The best result for each dataset is in **bold**.

| Method | WN-v1 | | WN-v2 | | WN-v3 | | WN-v4 | | FB-v1 | | FB-v2 | |
|---|---|---|---|---|---|---|---|---|---|---|---|---|
| | MRR | H@10 | MRR | H@10 | MRR | H@10 | MRR | H@10 | MRR | H@10 | MRR | H@10 |
| ULTRA(3KG) | 0.648 | 0.768 | 0.663 | 0.765 | 0.376 | 0.476 | 0.611 | 0.705 | **0.498** | **0.656** | **0.512** | **0.700** |
| HYPER(3KG) | **0.703** | **0.799** | 0.681 | **0.788** | **0.400** | **0.522** | **0.644** | **0.721** | 0.450 | 0.622 | 0.474 | 0.668 |
| HYPER(4HG) | 0.530 | 0.720 | 0.533 | 0.691 | 0.287 | 0.392 | 0.514 | 0.652 | 0.263 | 0.476 | 0.308 | 0.527 |
| HYPER(3KG + 2HG) | 0.702 | 0.782 | **0.686** | 0.785 | 0.385 | 0.503 | 0.640 | 0.710 | 0.454 | 0.648 | 0.480 | 0.695 |

| Method | FB-v3 | | FB-v4 | | NL-v1 | | NL-v2 | | NL-v3 | | NL-v4 | |
|---|---|---|---|---|---|---|---|---|---|---|---|---|
| | MRR | H@10 | MRR | H@10 | MRR | H@10 | MRR | H@10 | MRR | H@10 | MRR | H@10 |
| ULTRA(3KG) | **0.491** | **0.654** | **0.486** | **0.677** | **0.785** | **0.913** | **0.526** | 0.707 | **0.515** | 0.702 | 0.479 | 0.712 |
| HYPER(3KG) | 0.460 | 0.627 | 0.460 | 0.653 | 0.619 | 0.868 | 0.514 | 0.719 | 0.510 | 0.692 | 0.468 | 0.697 |
| HYPER(4HG) | 0.276 | 0.482 | 0.280 | 0.504 | 0.516 | 0.863 | 0.345 | 0.639 | 0.340 | 0.610 | 0.269 | 0.582 |
| HYPER(3KG + 2HG) | 0.466 | 0.648 | 0.460 | 0.663 | 0.570 | 0.719 | 0.521 | **0.741** | 0.509 | **0.705** | **0.501** | **0.728** |

| Method | ILPC Small | | ILPC Large | | HM 1k | | HM 3k | | HM 5k | | HM Indigo | |
|---|---|---|---|---|---|---|---|---|---|---|---|---|
| | MRR | H@10 | MRR | H@10 | MRR | H@10 | MRR | H@10 | MRR | H@10 | MRR | H@10 |
| ULTRA(3KG) | **0.302** | 0.443 | 0.290 | **0.424** | **0.059** | 0.092 | **0.037** | 0.077 | **0.034** | 0.071 | **0.440** | **0.648** |
| HYPER(3KG) | 0.291 | 0.438 | **0.293** | 0.412 | 0.046 | 0.092 | 0.036 | 0.073 | 0.033 | 0.069 | 0.437 | 0.644 |
| HYPER(4HG) | 0.169 | 0.347 | 0.183 | 0.327 | 0.027 | 0.075 | 0.024 | 0.064 | 0.024 | 0.058 | 0.298 | 0.484 |
| HYPER(3KG + 2HG) | 0.296 | **0.448** | 0.289 | 0.417 | 0.043 | **0.106** | **0.037** | **0.092** | **0.034** | **0.086** | 0.401 | 0.614 |

Table 10: Scalability comparison on FB15k-237 with batch size = 64.

| Model | # Parameters | Training Time (s/batch) | Inference Time (s/batch) | GPU Memory (GB) |
|---|---|---|---|---|
| ULTRA | 168,705 | 1.19 | 0.066 | 12.87 |
| HCNet | 159,297 | 2.64 | 0.156 | 18.03 |
| HYPER | 225,409 | 4.51 | 0.272 | 25.30 |

Table 11: Zero-shot MRR results on node and relation inductive knowledge hypergraph datasets. Superscript ‡ means the model is applied over the reification shown in the main body, and † means the model is applied with the alternative reification.

| Method | JF | | | | MFB | | | | WP | | | | WD | | | |
|---|---|---|---|---|---|---|---|---|---|---|---|---|---|---|---|---|
| | 25 | 50 | 75 | 100 | 25 | 50 | 75 | 100 | 25 | 50 | 75 | 100 | 25 | 50 | 75 | 100 |
| ULTRA‡(3KG) | 0.103 | 0.437 | 0.168 | 0.144 | 0.255 | 0.235 | 0.154 | 0.277 | 0.039 | 0.077 | 0.073 | 0.078 | 0.117 | 0.155 | 0.116 | 0.161 |
| ULTRA‡(4KG) | 0.011 | 0.298 | 0.042 | 0.082 | 0.217 | 0.170 | 0.043 | 0.135 | 0.009 | 0.006 | 0.006 | 0.004 | 0.027 | 0.069 | 0.063 | 0.065 |
| ULTRA‡(50KG) | 0.001 | 0.096 | 0.010 | 0.001 | 0.225 | 0.083 | 0.001 | 0.190 | 0.006 | 0.009 | 0.009 | 0.004 | 0.008 | 0.001 | 0.001 | 0.001 |
| ULTRA‡(4HG) | 0.154 | 0.442 | 0.175 | 0.170 | 0.338 | 0.236 | 0.129 | 0.280 | 0.052 | 0.091 | 0.089 | 0.089 | 0.076 | 0.175 | 0.052 | 0.136 |
| ULTRA‡(3KG+2HG) | 0.209 | 0.446 | 0.187 | 0.168 | 0.343 | 0.236 | 0.128 | 0.283 | 0.045 | 0.086 | 0.080 | 0.090 | 0.183 | 0.182 | 0.127 | 0.137 |
| ULTRA†(3KG) | 0.119 | 0.304 | 0.109 | 0.091 | 0.209 | 0.153 | 0.062 | 0.222 | 0.040 | 0.070 | 0.067 | 0.071 | 0.171 | **0.201** | 0.149 | 0.176 |
| ULTRA†(4KG) | 0.099 | 0.325 | 0.102 | 0.132 | 0.343 | 0.215 | 0.111 | 0.274 | 0.047 | 0.091 | 0.089 | 0.086 | 0.094 | 0.141 | 0.054 | 0.075 |
| ULTRA†(50KG) | 0.147 | 0.407 | 0.126 | 0.111 | 0.310 | 0.218 | 0.100 | 0.262 | 0.045 | 0.071 | 0.045 | 0.065 | 0.062 | 0.124 | 0.104 | 0.150 |
| ULTRA†(4HG) | 0.093 | 0.293 | 0.102 | 0.113 | 0.226 | 0.143 | 0.065 | 0.236 | 0.043 | 0.069 | 0.065 | 0.068 | 0.147 | 0.159 | 0.092 | 0.089 |
| ULTRA†(3KG+2HG) | 0.114 | 0.340 | 0.121 | 0.120 | 0.272 | 0.197 | 0.109 | 0.240 | 0.026 | 0.070 | 0.052 | 0.066 | 0.128 | 0.167 | 0.117 | 0.138 |
| HYPER(3KG) | 0.148 | 0.297 | 0.112 | 0.130 | 0.248 | 0.191 | 0.039 | 0.276 | **0.143** | 0.147 | 0.186 | 0.221 | 0.167 | 0.158 | 0.123 | 0.146 |
| HYPER(4KG) | 0.109 | 0.065 | 0.128 | 0.087 | 0.116 | 0.117 | 0.089 | 0.148 | 0.074 | 0.212 | 0.175 | 0.180 | 0.148 | 0.111 | 0.150 | 0.255 |
| HYPER(50KG) | 0.056 | 0.294 | 0.084 | 0.111 | 0.122 | 0.156 | 0.126 | 0.148 | 0.067 | 0.198 | 0.191 | 0.155 | 0.073 | 0.055 | 0.130 | 0.088 |
| HYPER(4HG) | 0.187 | 0.377 | 0.188 | **0.181** | 0.349 | 0.244 | 0.139 | 0.278 | 0.075 | 0.068 | 0.086 | 0.168 | 0.087 | 0.158 | 0.057 | 0.165 |
| HYPER(3KG+2HG) | **0.216** | **0.455** | **0.213** | 0.173 | **0.363** | **0.250** | **0.140** | **0.299** | 0.132 | **0.152** | **0.192** | **0.222** | **0.223** | 0.200 | **0.154** | **0.182** |

Table 12: Average zero-shot inference MRR over 16 newly proposed dataset comparison across KG-ICL, ULTRA, and HYPER variants.

| Model | Average MRR |
|---|---|
| KG-ICL(4 Layer) | 0.139 |
| KG-ICL(5 Layer) | 0.048 |
| KG-ICL(6 Layer) | 0.143 |
| ULTRA(3KG) | 0.162 |
| ULTRA(4KG) | 0.078 |
| ULTRA(50KG) | 0.040 |
| ULTRA(4HG) | 0.168 |
| ULTRA(3KG+2HG) | 0.183 |
| HYPER (3KG) | 0.161 |
| HYPER (4KG) | 0.135 |
| HYPER (50KG) | 0.135 |
| HYPER (4HG) | 0.128 |
| HYPER (3KG+2HG) | **0.236** |

Table 13: Training datasets for model variants

| Model | Knowledge Hypergraph | | | | Knowledge Graph | | | | |
|---|---|---|---|---|---|---|---|---|---|
| | JF17K | FB-AUTO | Wikipeople | MFB15K | FB15k-237 | WN18RR | CodEx Medium | NELL995 | Others(46G) |
| ULTRA(3KG) | | | | | ✓ | ✓ | ✓ | | |
| ULTRA(4KG) | | | | | ✓ | ✓ | ✓ | ✓ | |
| ULTRA(50KG) | | | | | ✓ | ✓ | ✓ | ✓ | ✓ |
| ULTRA(4HG) | ✓ | ✓ | ✓ | ✓ | | | | | |
| ULTRA(3KG + 2HG) | ✓ | | ✓ | | ✓ | ✓ | ✓ | | |
| HYPER(3KG) | | | | | ✓ | ✓ | ✓ | | |
| HYPER(4KG) | | | | | ✓ | ✓ | ✓ | ✓ | |
| HYPER(50KG) | | | | | ✓ | ✓ | ✓ | ✓ | ✓ |
| HYPER(4HG) | ✓ | ✓ | ✓ | ✓ | | | | | |
| HYPER(3KG + 2HG) | ✓ | | ✓ | | ✓ | ✓ | ✓ | | |
| HYPER(end2end) HCNet | Trained directly on target dataset's training graph | | | | | | | | |

Table 14: Arity distribution across node inductive datasets.

| Dataset | Arity | Training Graph | Inference Graph | Training % | Inference % |
|---------|-------|----------------|-----------------|------------|-------------|
| JF-IND | 2 | 264 | 9 | 4.28% | 2.93% |
| | 3 | 4586 | 216 | 74.36% | 70.36% |
| | 4 | 1317 | 82 | 21.36% | 26.71% |
| | ≥5 | 0 | 0 | 0.00% | 0.00% |
| WP-IND | 2 | 0 | 0 | 0.00% | 0.00% |
| | 3 | 3375 | 476 | 81.54% | 86.39% |
| | 4 | 764 | 75 | 18.46% | 13.61% |
| | ≥5 | 0 | 0 | 0.00% | 0.00% |
| MFB-IND | 2 | 0 | 0 | 0.00% | 0.00% |
| | 3 | 336733 | 7527 | 100.00% | 100.00% |
| | 4 | 0 | 0 | 0.00% | 0.00% |
| | ≥5 | 0 | 0 | 0.00% | 0.00% |

Table 15: Dataset statistics of inductive link prediction task with knowledge hypergraph.

| Statistic | JF-IND | WP-IND | MFB-IND |
|-----------|--------|--------|---------|
| # seen vertices | 4,685 | 4,463 | 3,283 |
| # train hyperedges | 6,167 | 4,139 | 336,733 |
| # unseen vertices | 100 | 100 | 500 |
| # relations | 31 | 32 | 12 |
| # max arity | 4 | 4 | 3 |

Table 16: Dataset statistics of pretrained knowledge hypergraphs and knowledge graphs with respective arity.

| Dataset | FB-AUTO | WikiPeople | JF17K | MFB15K | FB15k237 | WN18RR | CoDEx-M |
|---------|---------|------------|-------|--------|----------|--------|---------|
| $|V|$ | 3,410 | 47,765 | 29,177 | 10,314 | 14541 | 40943 | 17050 |
| $|R|$ | 8 | 707 | 327 | 71 | 237 | 11 | 51 |
| # train | 6,778 | 305,725 | 61,104 | 415,375 | 272115 | 86835 | 185584 |
| # valid | 2,255 | 38,223 | 15,275 | 39,348 | 17535 | 3034 | 10310 |
| # test | 2,180 | 38,281 | 24,915 | 38,797 | 20466 | 3134 | 10311 |
| # max arity | 5 | 9 | 6 | 5 | 2 | 2 | 2 |
| # arity= 2 | 3,786 | 337,914 | 56,322 | 82,247 | 310,116 | 93,003 | 206,205 |
| # arity= 3 | 0 | 25,820 | 34,550 | 400,027 | 0 | 0 | 0 |
| # arity= 4 | 215 | 15,188 | 9,509 | 26 | 0 | 0 | 0 |
| # arity≥ 5 | 7,212 | 3,307 | 2,267 | 11,220 | 0 | 0 | 0 |

Table 17: Dataset statistics for inductive on both node and relation link prediction datasets. Triples are the number of edges given at training, validation, or test graphs, respectively, whereas Valid and Test denote triples to be predicted in the validation and test graphs.

| Dataset | Training Graph | | | Validation Graph | | | | Test Graph | | | |
|---|---|---|---|---|---|---|---|---|---|---|---|
| | Entities | Rels | Triples | Entities | Rels | Triples | Valid | Entities | Rels | Triples | Test |
| FB-25 | 5190 | 163 | 91571 | 4097 | 216 | 17147 | 5716 | 4097 | 216 | 17147 | 5716 |
| FB-50 | 5190 | 153 | 85375 | 4445 | 205 | 11636 | 3879 | 4445 | 205 | 11636 | 3879 |
| FB-75 | 4659 | 134 | 62809 | 2792 | 186 | 9316 | 3106 | 2792 | 186 | 9316 | 3106 |
| FB-100 | 4659 | 134 | 62809 | 2624 | 77 | 6987 | 2329 | 2624 | 77 | 6987 | 2329 |
| WK-25 | 12659 | 47 | 41873 | 3228 | 74 | 3391 | 1130 | 3228 | 74 | 3391 | 1131 |
| WK-50 | 12022 | 72 | 82481 | 9328 | 93 | 9672 | 3224 | 9328 | 93 | 9672 | 3225 |
| WK-75 | 6853 | 52 | 28741 | 2722 | 65 | 3430 | 1143 | 2722 | 65 | 3430 | 1144 |
| WK-100 | 9784 | 67 | 49875 | 12136 | 37 | 13487 | 4496 | 12136 | 37 | 13487 | 4496 |
| NL-0 | 1814 | 134 | 7796 | 2026 | 112 | 2287 | 763 | 2026 | 112 | 2287 | 763 |
| NL-25 | 4396 | 106 | 17578 | 2146 | 120 | 2230 | 743 | 2146 | 120 | 2230 | 744 |
| NL-50 | 4396 | 106 | 17578 | 2335 | 119 | 2576 | 859 | 2335 | 119 | 2576 | 859 |
| NL-75 | 2607 | 96 | 11058 | 1578 | 116 | 1818 | 606 | 1578 | 116 | 1818 | 607 |
| NL-100 | 1258 | 55 | 7832 | 1709 | 53 | 2378 | 793 | 1709 | 53 | 2378 | 793 |
| Metafam | 1316 | 28 | 13821 | 1316 | 28 | 13821 | 590 | 656 | 28 | 7257 | 184 |
| FBNELL | 4636 | 100 | 10275 | 4636 | 100 | 10275 | 1055 | 4752 | 183 | 10685 | 597 |
| Wiki MT1 tax | 10000 | 10 | 17178 | 10000 | 10 | 17178 | 1908 | 10000 | 9 | 16526 | 1834 |
| Wiki MT1 health | 10000 | 7 | 14371 | 10000 | 7 | 14371 | 1596 | 10000 | 7 | 14110 | 1566 |
| Wiki MT2 org | 10000 | 10 | 23233 | 10000 | 10 | 23233 | 2581 | 10000 | 11 | 21976 | 2441 |
| Wiki MT2 sci | 10000 | 16 | 16471 | 10000 | 16 | 16471 | 1830 | 10000 | 16 | 14852 | 1650 |
| Wiki MT3 art | 10000 | 45 | 27262 | 10000 | 45 | 27262 | 3026 | 10000 | 45 | 28023 | 3113 |
| Wiki MT3 infra | 10000 | 24 | 21990 | 10000 | 24 | 21990 | 2443 | 10000 | 27 | 21646 | 2405 |
| Wiki MT4 sci | 10000 | 42 | 12576 | 10000 | 42 | 12576 | 1397 | 10000 | 42 | 12516 | 1388 |
| Wiki MT4 health | 10000 | 21 | 15539 | 10000 | 21 | 15539 | 1725 | 10000 | 20 | 15337 | 1703 |

Table 18: Dataset statistics for inductive-$e$ link prediction datasets. Triples are the number of edges given at training, validation, or test graphs, respectively, whereas Valid and Test denote triples to be predicted in the validation and test graphs.

| Dataset | Rels | Training Graph | | Validation Graph | | | Test Graph | | |
|---|---|---|---|---|---|---|---|---|---|
| | | Entities | Triples | Entities | Triples | Valid | Entities | Triples | Test |
| FB-v1 | 180 | 1594 | 4245 | 1594 | 4245 | 489 | 1093 | 1993 | 411 |
| FB-v2 | 200 | 2608 | 9739 | 2608 | 9739 | 1166 | 1660 | 4145 | 947 |
| FB-v3 | 215 | 3668 | 17986 | 3668 | 17986 | 2194 | 2501 | 7406 | 1731 |
| FB-v4 | 219 | 4707 | 27203 | 4707 | 27203 | 3352 | 3051 | 11714 | 2840 |
| WN-v1 | 9 | 2746 | 5410 | 2746 | 5410 | 630 | 922 | 1618 | 373 |
| WN-v2 | 10 | 6954 | 15262 | 6954 | 15262 | 1838 | 2757 | 4011 | 852 |
| WN-v3 | 11 | 12078 | 25901 | 12078 | 25901 | 3097 | 5084 | 6327 | 1143 |
| WN-v4 | 9 | 3861 | 7940 | 3861 | 7940 | 934 | 7084 | 12334 | 2823 |
| NL-v1 | 14 | 3103 | 4687 | 3103 | 4687 | 414 | 225 | 833 | 201 |
| NL-v2 | 88 | 2564 | 8219 | 2564 | 8219 | 922 | 2086 | 4586 | 935 |
| NL-v3 | 142 | 4647 | 16393 | 4647 | 16393 | 1851 | 3566 | 8048 | 1620 |
| NL-v4 | 76 | 2092 | 7546 | 2092 | 7546 | 876 | 2795 | 7073 | 1447 |
| ILPC Small | 48 | 10230 | 78616 | 6653 | 20960 | 2908 | 6653 | 20960 | 2902 |
| ILPC Large | 65 | 46626 | 202446 | 29246 | 77044 | 10179 | 29246 | 77044 | 10184 |
| HM 1k | 11 | 36237 | 93364 | 36311 | 93364 | 1771 | 9899 | 18638 | 476 |
| HM 3k | 11 | 32118 | 71097 | 32250 | 71097 | 1201 | 19218 | 38285 | 1349 |
| HM 5k | 11 | 28601 | 57601 | 28744 | 57601 | 900 | 23792 | 48425 | 2124 |
| HM Indigo | 229 | 12721 | 121601 | 12797 | 121601 | 14121 | 14775 | 250195 | 14904 |

Table 19: Experiment result on node and relation inductive knowledge hypergraph datasets for JF and WP.

| Method | JF-25 | | | | JF-50 | | | | JF-75 | | | | JF-100 | | | |
|---|---|---|---|---|---|---|---|---|---|---|---|---|---|---|---|---|
| | MRR | H@1 | H@3 | H@10 | MRR | H@1 | H@3 | H@10 | MRR | H@1 | H@3 | H@10 | MRR | H@1 | H@3 | H@10 |
| G-MPNN | 0.006 | 0.004 | 0.004 | 0.007 | 0.003 | 0.000 | 0.000 | 0.003 | 0.001 | 0.000 | 0.000 | 0.000 | 0.002 | 0.000 | 0.000 | 0.000 |
| HCNet | 0.011 | 0.004 | 0.007 | 0.011 | 0.009 | 0.000 | 0.000 | 0.024 | 0.069 | 0.038 | 0.072 | 0.125 | 0.028 | 0.000 | 0.018 | 0.054 |
| HYPER(end2end) | 0.202 | 0.117 | 0.226 | 0.346 | **0.468** | **0.358** | 0.540 | 0.653 | 0.207 | **0.125** | 0.226 | 0.357 | **0.198** | **0.107** | 0.161 | 0.411 |
| | ±.003 | ±.002 | ±.006 | ±.005 | ±.004 | ±.007 | ±.002 | ±.003 | ±.005 | ±.004 | ±.006 | ±.003 | ±.008 | ±.002 | ±.004 | ±.003 |
| ULTRA[‡](3KG)(0-shot) | 0.103 | 0.039 | 0.095 | 0.240 | 0.437 | 0.301 | 0.581 | 0.699 | 0.168 | 0.100 | 0.160 | 0.288 | 0.144 | 0.089 | 0.143 | 0.196 |
| ULTRA[‡](4KG)(0-shot) | 0.011 | 0.004 | 0.011 | 0.035 | 0.298 | 0.196 | 0.368 | 0.468 | 0.042 | 0.025 | 0.050 | 0.072 | 0.082 | 0.071 | 0.089 | 0.107 |
| ULTRA[‡](50KG)(0-shot) | 0.001 | 0.000 | 0.000 | 0.000 | 0.096 | 0.089 | 0.102 | 0.105 | 0.010 | 0.006 | 0.013 | 0.016 | 0.001 | 0.000 | 0.000 | 0.000 |
| ULTRA[‡](4HG)(0-shot) | 0.154 | 0.067 | 0.155 | 0.336 | 0.442 | 0.288 | 0.539 | 0.696 | 0.175 | 0.085 | 0.185 | 0.339 | 0.170 | **0.107** | 0.143 | 0.286 |
| ULTRA[‡](3KG+2HG)(0-shot) | 0.209 | 0.113 | 0.216 | **0.413** | 0.446 | 0.293 | 0.547 | **0.707** | 0.187 | 0.097 | 0.197 | 0.342 | 0.168 | **0.107** | 0.143 | 0.286 |
| HYPER(3KG)(0-shot) | 0.148 | 0.071 | 0.152 | 0.318 | 0.297 | 0.212 | 0.336 | 0.460 | 0.112 | 0.041 | 0.132 | 0.254 | 0.130 | 0.018 | 0.107 | 0.375 |
| HYPER(4KG)(0-shot) | 0.109 | 0.078 | 0.122 | 0.178 | 0.065 | 0.009 | 0.037 | 0.241 | 0.128 | 0.078 | 0.104 | 0.247 | 0.087 | 0.000 | 0.054 | 0.286 |
| HYPER(50KG)(0-shot) | 0.056 | 0.022 | 0.044 | 0.167 | 0.294 | 0.204 | 0.352 | 0.463 | 0.084 | 0.026 | 0.104 | 0.208 | 0.111 | 0.018 | 0.089 | 0.321 |
| HYPER(4HG)(0-shot) | 0.187 | 0.095 | 0.219 | 0.360 | 0.377 | 0.239 | 0.476 | 0.608 | 0.188 | 0.110 | 0.204 | **0.370** | 0.181 | 0.089 | 0.161 | **0.464** |
| HYPER(3KG + 2HG)(0-shot) | 0.216 | 0.122 | **0.233** | **0.413** | 0.455 | 0.325 | **0.556** | 0.664 | **0.213** | 0.122 | 0.231 | 0.367 | 0.173 | 0.071 | **0.179** | 0.446 |
| ULTRA[‡](3KG+2HG)(finetuned) | 0.214 | 0.109 | 0.221 | 0.406 | 0.438 | 0.301 | 0.539 | 0.698 | 0.193 | 0.102 | 0.204 | 0.351 | 0.174 | **0.103** | 0.149 | 0.279 |
| | ±.005 | ±.003 | ±.006 | ±.007 | ±.006 | ±.004 | ±.008 | ±.007 | ±.003 | ±.002 | ±.005 | ±.006 | ±.004 | ±.003 | ±.002 | ±.005 |
| HYPER(3KG + 2HG)(finetuned) | **0.217** | **0.131** | 0.226 | 0.389 | 0.456 | 0.331 | 0.554 | 0.672 | 0.209 | 0.119 | **0.238** | 0.361 | 0.176 | 0.089 | 0.161 | 0.393 |
| | ±.001 | ±.002 | ±.004 | ±.006 | ±.003 | ±.005 | ±.002 | ±.001 | ±.004 | ±.006 | ±.003 | ±.007 | ±.005 | ±.003 | ±.008 | ±.002 |

| Method | WP-25 | | | | WP-50 | | | | WP-75 | | | | WP-100 | | | |
|---|---|---|---|---|---|---|---|---|---|---|---|---|---|---|---|---|
| | MRR | H@1 | H@3 | H@10 | MRR | H@1 | H@3 | H@10 | MRR | H@1 | H@3 | H@10 | MRR | H@1 | H@3 | H@10 |
| G-MPNN | 0.005 | 0.003 | 0.005 | 0.006 | 0.002 | 0.001 | 0.000 | 0.002 | 0.001 | 0.000 | 0.001 | 0.000 | 0.000 | 0.000 | 0.000 | 0.001 |
| HCNet | 0.104 | 0.048 | 0.114 | 0.230 | 0.050 | 0.025 | 0.059 | 0.087 | 0.019 | 0.010 | 0.020 | 0.032 | 0.003 | 0.000 | 0.000 | 0.003 |
| HYPER(end2end) | 0.159 | 0.071 | **0.172** | 0.358 | 0.143 | 0.082 | 0.157 | 0.260 | 0.139 | 0.072 | 0.129 | 0.294 | 0.202 | 0.106 | 0.190 | 0.418 |
| | ±.003 | ±.004 | ±.005 | ±.006 | ±.002 | ±.003 | ±.004 | ±.007 | ±.003 | ±.001 | ±.005 | ±.006 | ±.002 | ±.003 | ±.004 | ±.007 |
| ULTRA[‡](3KG)(0-shot) | 0.039 | 0.015 | 0.034 | 0.108 | 0.077 | 0.035 | 0.098 | 0.160 | 0.073 | 0.037 | 0.079 | 0.158 | 0.078 | 0.048 | 0.084 | 0.148 |
| ULTRA[‡](4KG)(0-shot) | 0.009 | 0.000 | 0.002 | 0.041 | 0.006 | 0.000 | 0.002 | 0.022 | 0.006 | 0.001 | 0.001 | 0.024 | 0.004 | 0.000 | 0.000 | 0.023 |
| ULTRA[‡](50KG)(0-shot) | 0.006 | 0.005 | 0.007 | 0.007 | 0.009 | 0.007 | 0.012 | 0.012 | 0.009 | 0.008 | 0.010 | 0.011 | 0.004 | 0.003 | 0.006 | 0.006 |
| ULTRA[‡](4HG)(0-shot) | 0.052 | 0.023 | 0.070 | 0.103 | 0.091 | 0.044 | 0.121 | 0.166 | 0.089 | 0.048 | 0.102 | 0.168 | 0.089 | 0.055 | 0.103 | 0.158 |
| ULTRA[‡](3KG+2HG)(0-shot) | 0.045 | 0.016 | 0.060 | 0.096 | 0.086 | 0.041 | 0.110 | 0.161 | 0.080 | 0.039 | 0.096 | 0.159 | 0.090 | 0.055 | 0.109 | 0.158 |
| HYPER(3KG)(0-shot) | 0.143 | 0.057 | 0.138 | 0.349 | 0.147 | 0.073 | 0.159 | **0.327** | 0.186 | 0.097 | 0.186 | **0.391** | 0.221 | 0.106 | **0.241** | **0.498** |
| HYPER(4KG)(0-shot) | 0.074 | 0.016 | 0.094 | 0.219 | 0.212 | 0.141 | 0.250 | 0.391 | 0.175 | 0.078 | 0.188 | 0.438 | 0.180 | 0.062 | 0.188 | 0.516 |
| HYPER(50KG)(0-shot) | 0.067 | 0.016 | 0.094 | 0.156 | 0.198 | 0.109 | 0.234 | 0.406 | 0.191 | 0.078 | 0.172 | 0.500 | 0.155 | 0.031 | 0.109 | 0.563 |
| HYPER(4HG)(0-shot) | 0.075 | 0.033 | 0.094 | 0.186 | 0.068 | 0.056 | 0.080 | 0.081 | 0.086 | 0.066 | 0.107 | 0.114 | 0.168 | 0.080 | 0.190 | 0.360 |
| HYPER(3KG + 2HG)(0-shot) | 0.132 | 0.058 | 0.151 | 0.296 | 0.152 | 0.086 | 0.178 | 0.295 | 0.192 | 0.107 | **0.201** | 0.384 | **0.222** | **0.132** | 0.209 | 0.453 |
| ULTRA[‡](3KG+2HG)(finetuned) | 0.051 | 0.019 | 0.066 | 0.104 | 0.092 | 0.046 | 0.118 | 0.169 | 0.086 | 0.044 | 0.103 | 0.167 | 0.097 | 0.061 | 0.116 | 0.166 |
| | ±.003 | ±.002 | ±.004 | ±.005 | ±.004 | ±.003 | ±.006 | ±.007 | ±.005 | ±.002 | ±.004 | ±.006 | ±.004 | ±.003 | ±.005 | ±.007 |
| HYPER(3KG + 2HG)(finetuned) | **0.169** | **0.078** | 0.164 | **0.399** | **0.171** | **0.103** | **0.201** | 0.306 | **0.194** | **0.112** | 0.199 | 0.375 | 0.210 | 0.116 | 0.206 | 0.424 |
| | ±.003 | ±.002 | ±.004 | ±.005 | ±.001 | ±.006 | ±.002 | ±.007 | ±.003 | ±.004 | ±.002 | ±.005 | ±.006 | ±.003 | ±.004 | ±.002 |

Table 20: Experiment result on node and relation inductive knowledge hypergraph datasets for WD and MFB.

| Method | WD-25 | | | | WD-50 | | | | WD-75 | | | | WD-100 | | | |
|---|---|---|---|---|---|---|---|---|---|---|---|---|---|---|---|---|
| | MRR | H@1 | H@3 | H@10 | MRR | H@1 | H@3 | H@10 | MRR | H@1 | H@3 | H@10 | MRR | H@1 | H@3 | H@10 |
| G-MPNN | 0.001 | 0.000 | 0.000 | 0.000 | 0.001 | 0.000 | 0.000 | 0.000 | 0.001 | 0.000 | 0.000 | 0.001 | 0.001 | 0.000 | 0.000 | 0.001 |
| HCNet | 0.086 | 0.050 | 0.096 | 0.136 | 0.043 | 0.027 | 0.044 | 0.060 | 0.015 | 0.006 | 0.023 | 0.030 | 0.007 | 0.004 | 0.005 | 0.007 |
| HYPER(end2end) | 0.215 | 0.132 | 0.225 | 0.394 | 0.205 | 0.153 | 0.219 | 0.317 | **0.172** | **0.105** | **0.194** | 0.298 | 0.205 | 0.139 | 0.226 | 0.342 |
| | ±.002 | ±.006 | ±.005 | ±.004 | ±.007 | ±.003 | ±.006 | ±.002 | ±.003 | ±.004 | ±.003 | ±.007 | ±.002 | ±.008 | ±.004 | ±.005 |
| ULTRA‡(3KG)(0-shot) | 0.117 | 0.040 | 0.132 | 0.315 | 0.155 | 0.087 | 0.175 | 0.311 | 0.116 | 0.060 | 0.109 | 0.263 | 0.161 | 0.105 | 0.165 | 0.311 |
| ULTRA‡(4KG)(0-shot) | 0.027 | 0.013 | 0.033 | 0.063 | 0.069 | 0.060 | 0.077 | 0.093 | 0.063 | 0.040 | 0.069 | 0.128 | 0.065 | 0.040 | 0.070 | 0.137 |
| ULTRA‡(50KG)(0-shot) | 0.008 | 0.007 | 0.010 | 0.010 | 0.001 | 0.000 | 0.000 | 0.000 | 0.001 | 0.000 | 0.000 | 0.000 | 0.001 | 0.000 | 0.004 | 0.004 |
| ULTRA‡(4HG)(0-shot) | 0.076 | 0.096 | 0.185 | 0.348 | 0.175 | 0.109 | 0.191 | 0.311 | 0.052 | 0.084 | 0.162 | 0.303 | 0.136 | 0.128 | 0.212 | 0.344 |
| ULTRA‡(3KG+2HG)(0-shot) | 0.183 | 0.103 | 0.209 | 0.351 | 0.182 | 0.115 | 0.202 | 0.328 | 0.127 | 0.092 | 0.161 | 0.302 | 0.137 | 0.130 | 0.211 | 0.347 |
| HYPER(3KG)(0-shot) | 0.167 | 0.010 | 0.172 | 0.331 | 0.158 | 0.104 | 0.175 | 0.295 | 0.123 | 0.005 | 0.142 | 0.255 | 0.146 | 0.077 | 0.168 | 0.281 |
| HYPER(4KG)(0-shot) | 0.148 | 0.061 | 0.167 | 0.394 | 0.111 | 0.016 | 0.203 | 0.297 | 0.150 | 0.047 | 0.188 | 0.375 | 0.255 | 0.209 | 0.269 | 0.373 |
| HYPER(50KG)(0-shot) | 0.073 | 0.046 | 0.091 | 0.136 | 0.055 | 0.031 | 0.078 | 0.109 | 0.130 | 0.078 | 0.141 | 0.281 | 0.088 | 0.015 | 0.164 | 0.194 |
| HYPER(4HG)(0-shot) | 0.087 | 0.076 | 0.093 | 0.103 | 0.158 | 0.142 | 0.164 | 0.197 | 0.057 | 0.051 | 0.062 | 0.064 | 0.165 | 0.139 | 0.177 | 0.233 |
| HYPER(3KG + 2HG)(0-shot) | 0.223 | **0.156** | 0.225 | **0.404** | 0.200 | 0.148 | 0.208 | 0.317 | 0.154 | 0.093 | 0.168 | 0.275 | 0.182 | 0.133 | 0.177 | 0.286 |
| ULTRA‡(3KG+2HG)(finetuned) | 0.191 | 0.108 | 0.218 | 0.364 | 0.189 | 0.121 | 0.210 | 0.341 | 0.134 | 0.097 | 0.169 | **0.314** | 0.145 | 0.137 | 0.219 | 0.349 |
| | ±.004 | ±.003 | ±.005 | ±.006 | ±.003 | ±.004 | ±.006 | ±.007 | ±.004 | ±.002 | ±.005 | ±.006 | ±.003 | ±.004 | ±.006 | ±.007 |
| HYPER(3KG + 2HG)(finetuned) | **0.225** | 0.146 | **0.245** | 0.397 | **0.234** | **0.186** | **0.230** | **0.355** | 0.166 | 0.101 | 0.189 | 0.294 | **0.210** | **0.140** | **0.235** | **0.351** |
| | ±.003 | ±.005 | ±.004 | ±.006 | ±.002 | ±.007 | ±.003 | ±.001 | ±.005 | ±.003 | ±.006 | ±.004 | ±.002 | ±.008 | ±.003 | ±.005 |

| Method | MFB-25 | | | | MFB-50 | | | | MFB-75 | | | | MFB-100 | | | |
|---|---|---|---|---|---|---|---|---|---|---|---|---|---|---|---|---|
| | MRR | H@1 | H@3 | H@10 | MRR | H@1 | H@3 | H@10 | MRR | H@1 | H@3 | H@10 | MRR | H@1 | H@3 | H@10 |
| G-MPNN | 0.002 | 0.000 | 0.000 | 0.004 | 0.004 | 0.001 | 0.003 | 0.007 | 0.007 | 0.003 | 0.004 | 0.010 | 0.003 | 0.000 | 0.002 | 0.005 |
| HCNet | 0.033 | 0.008 | 0.008 | 0.108 | 0.026 | 0.013 | 0.022 | 0.041 | 0.016 | 0.007 | 0.009 | 0.021 | 0.082 | 0.028 | 0.085 | 0.227 |
| HYPER(end2end) | 0.332 | 0.221 | 0.388 | 0.533 | 0.200 | 0.105 | 0.251 | 0.374 | 0.135 | 0.070 | 0.143 | 0.255 | 0.222 | 0.169 | 0.228 | 0.317 |
| | ±.005 | ±.004 | ±.003 | ±.002 | ±.006 | ±.004 | ±.007 | ±.005 | ±.003 | ±.006 | ±.002 | ±.008 | ±.001 | ±.005 | ±.004 | ±.003 |
| ULTRA‡(3KG)(0-shot) | 0.255 | 0.258 | 0.408 | 0.525 | 0.235 | 0.148 | 0.269 | 0.384 | 0.154 | 0.084 | 0.166 | 0.274 | 0.277 | 0.201 | 0.315 | 0.412 |
| ULTRA‡(4KG)(0-shot) | 0.217 | 0.158 | 0.267 | 0.308 | 0.170 | 0.113 | 0.206 | 0.269 | 0.043 | 0.025 | 0.048 | 0.084 | 0.135 | 0.032 | 0.234 | 0.271 |
| ULTRA‡(50KG)(0-shot) | 0.225 | 0.196 | 0.254 | 0.263 | 0.083 | 0.074 | 0.094 | 0.097 | 0.001 | 0.000 | 0.000 | 0.000 | 0.190 | 0.156 | 0.221 | 0.247 |
| ULTRA‡(4HG)(0-shot) | 0.338 | 0.225 | 0.400 | 0.521 | 0.236 | 0.145 | 0.282 | 0.404 | 0.129 | 0.081 | 0.171 | 0.344 | 0.280 | 0.194 | 0.323 | 0.435 |
| ULTRA‡(3KG+2HG)(0-shot) | 0.343 | 0.233 | 0.413 | 0.538 | 0.236 | 0.137 | 0.286 | **0.408** | 0.128 | 0.076 | **0.183** | 0.352 | 0.283 | 0.200 | 0.325 | 0.437 |
| HYPER(3KG)(0-shot) | 0.248 | 0.167 | 0.283 | 0.396 | 0.191 | 0.123 | 0.216 | 0.296 | 0.039 | 0.016 | 0.029 | 0.073 | 0.276 | 0.198 | 0.311 | 0.416 |
| HYPER(4KG)(0-shot) | 0.116 | 0.061 | 0.134 | 0.268 | 0.117 | 0.060 | 0.155 | 0.238 | 0.089 | 0.011 | 0.129 | 0.204 | 0.148 | 0.073 | 0.171 | 0.293 |
| HYPER(50KG)(0-shot) | 0.122 | 0.061 | 0.134 | 0.232 | 0.156 | 0.119 | 0.155 | 0.214 | 0.126 | 0.086 | 0.118 | 0.204 | 0.148 | 0.073 | 0.195 | 0.256 |
| HYPER(4HG)(0-shot) | 0.349 | 0.258 | 0.400 | 0.546 | 0.244 | **0.169** | 0.286 | 0.382 | 0.139 | 0.082 | 0.140 | 0.244 | 0.278 | 0.195 | 0.316 | 0.441 |
| HYPER(3KG + 2HG)(0-shot) | **0.363** | **0.263** | **0.417** | **0.550** | **0.250** | 0.167 | **0.287** | 0.393 | 0.140 | 0.077 | 0.140 | 0.260 | **0.299** | **0.214** | **0.339** | 0.449 |
| ULTRA‡(3KG+2HG)(finetuned) | 0.351 | 0.241 | 0.425 | 0.552 | 0.244 | 0.145 | 0.297 | 0.401 | 0.136 | 0.081 | 0.182 | 0.364 | 0.291 | 0.209 | 0.336 | 0.449 |
| | ±.005 | ±.004 | ±.006 | ±.007 | ±.003 | ±.002 | ±.005 | ±.006 | ±.003 | ±.002 | ±.004 | ±.006 | ±.004 | ±.003 | ±.005 | ±.007 |
| HYPER(3KG + 2HG)(finetuned) | 0.347 | 0.229 | 0.408 | 0.533 | 0.243 | 0.163 | 0.286 | 0.391 | **0.158** | **0.088** | 0.161 | **0.302** | 0.275 | 0.197 | 0.290 | **0.452** |
| | ±.004 | ±.006 | ±.003 | ±.002 | ±.005 | ±.006 | ±.007 | ±.004 | ±.002 | ±.005 | ±.003 | ±.006 | ±.002 | ±.008 | ±.003 | ±.004 |

Table 21: Experiment result on node-inductive knowledge hypergraph datasets.

| Method | JF-IND | | | WP-IND | | | MFB-IND | | |
|---|---|---|---|---|---|---|---|---|---|
| | MRR | H@1 | H@3 | MRR | H@1 | H@3 | MRR | H@1 | H@3 |
| HGNN | 0.102 | 0.086 | 0.128 | 0.072 | 0.045 | 0.112 | 0.121 | 0.076 | 0.114 |
| HyperGCN | 0.099 | 0.088 | 0.133 | 0.075 | 0.049 | 0.111 | 0.118 | 0.074 | 0.117 |
| G-MPNN | 0.219 | 0.155 | 0.236 | 0.177 | 0.108 | 0.191 | 0.124 | 0.071 | 0.123 |
| RD-MPNN | 0.402 | 0.308 | 0.453 | 0.304 | 0.238 | 0.328 | 0.122 | 0.082 | 0.125 |
| HCNet | 0.435 | 0.357 | 0.495 | 0.414 | 0.352 | 0.451 | 0.368 | 0.223 | 0.417 |
| HYPER(end2end) | 0.422 | 0.320 | 0.483 | 0.435 | 0.367 | 0.471 | 0.427 | 0.290 | 0.499 |
| | ±.004 | ±.006 | ±.007 | ±.005 | ±.004 | ±.006 | ±.003 | ±.005 | ±.004 |
| ULTRA$^{\ddagger}$(3KG)(0-shot) | 0.321 | 0.221 | 0.383 | 0.305 | 0.242 | 0.320 | 0.277 | 0.161 | 0.316 |
| ULTRA$^{\ddagger}$(4KG)(0-shot) | 0.065 | 0.004 | 0.079 | 0.123 | 0.102 | 0.114 | 0.096 | 0.068 | 0.112 |
| ULTRA$^{\ddagger}$(50KG)(0-shot) | 0.008 | 0.000 | 0.002 | 0.029 | 0.024 | 0.027 | 0.026 | 0.020 | 0.031 |
| ULTRA$^{\ddagger}$(4HG)(0-shot) | 0.397 | 0.274 | 0.466 | 0.319 | 0.330 | 0.467 | 0.264 | 0.141 | 0.299 |
| ULTRA$^{\ddagger}$(3KG+2HG)(0-shot) | 0.410 | 0.294 | 0.489 | 0.341 | 0.352 | 0.480 | 0.294 | 0.164 | 0.325 |
| HYPER(3KG)(0-shot) | 0.263 | 0.177 | 0.281 | 0.259 | 0.176 | 0.307 | 0.184 | 0.123 | 0.196 |
| HYPER(4KG)(0-shot) | 0.266 | 0.182 | 0.313 | 0.231 | 0.168 | 0.267 | 0.120 | 0.073 | 0.115 |
| HYPER(50KG)(0-shot) | 0.302 | 0.212 | 0.354 | 0.253 | 0.188 | 0.248 | 0.248 | 0.157 | 0.271 |
| HYPER(4HG)(0-shot) | 0.403 | 0.277 | 0.501 | 0.375 | 0.297 | 0.410 | **0.497** | **0.351** | **0.582** |
| HYPER(3KG + 2HG)(0-shot) | 0.459 | 0.365 | 0.515 | 0.415 | 0.338 | 0.454 | 0.404 | 0.267 | 0.480 |
| ULTRA$^{\ddagger}$(3KG+2HG)(finetuned) | 0.421 | 0.302 | 0.501 | 0.349 | 0.361 | 0.492 | 0.303 | 0.171 | 0.337 |
| | ±.006 | ±.004 | ±.007 | ±.003 | ±.005 | ±.006 | ±.004 | ±.002 | ±.005 |
| HYPER(3KG + 2HG)(finetuned) | **0.463** | **0.373** | **0.517** | **0.446** | **0.379** | **0.482** | 0.455 | 0.318 | 0.530 |
| | ±.002 | ±.003 | ±.008 | ±.008 | ±.009 | ±.007 | ±.003 | ±.007 | ±.005 |

Table 22: HYPER hyper-parameters for pretraining, fine-tuning, and end-to-end training.

| | Hyperparameter | HYPER |
|---|---|---|
| Positional Interaction Encoder | # Layers | 2 |
| | Hidden dimension | 64 |
| | Dropout | 0 |
| | Activation | ReLU |
| Relation Encoder | # Layers $T$ | 6 |
| | Hidden dimension | 64 |
| | Dropout | 0 |
| | Activation | ReLU |
| | Norm | LayerNorm |
| Entity Encoder | # Layers $L$ | 6 |
| | Hidden dimension | 64 |
| | Dec | 2-layer MLP |
| | Dropout | 0 |
| | Activation | ReLU |
| | Norm | LayerNorm |
| Pre-training | Optimizer | AdamW |
| | Learning rate | 0.0005 |
| | Training steps | 30,000 |
| | Adversarial temperature | 1 |
| | # Negatives | 512 |
| | Batch size | 32 |
| Fine-tuning | Optimizer | AdamW |
| | Learning rate | 0.0005 |
| | Adversarial temperature | 1 |
| | # Negatives | 256 |
| | Batch size | 8 |
| End-to-End | Optimizer | AdamW |
| | Learning rate | 0.0005 |
| | Adversarial temperature | 1 |
| | # Negatives | 256 |
| | Batch size | 8 |

Table 23: Hyperparameters for fine-tuning and training end-to-end for HYPER.

| Datasets | Finetune | | End-to-End | |
|---|---|---|---|---|
| | Epoch | Batch per Epoch | Epoch | Batch per Epoch |
| JF 25-100 | 3 | full | 10 | full |
| WP 25-100 | 3 | full | 10 | full |
| MFB 25-100 | 3 | full | 10 | full |
| WD 25-100 | 3 | full | 10 | full |
| JF-IND | 1 | full | 20 | full |
| WP-IND | 1 | full | 20 | full |
| MFB-IND | 1 | 2000 | 4 | 10000 |
| FB 25-100 | 3 | full | 10 | full |
| WK 25-100 | 3 | full | 10 | full |
| NL 0-100 | 3 | full | 10 | full |
| MT1-MT4 | 3 | full | 10 | full |
| Metafam, FBNELL | 3 | full | 10 | full |
| FB v1-v4 | 1 | full | 10 | full |
| WN v1-v4 | 1 | full | 10 | full |
| NL v1-v4 | 3 | full | 10 | full |
| ILPC Small | 3 | full | 10 | full |
| ILPC Large | 1 | 1000 | 10 | 1000 |
| HM 1k-5k, Indigo | 1 | 100 | 10 | 1000 |

