# OpenReview forum: "HYPER: A Foundation Model for Inductive Link Prediction with Knowledge Hypergraphs"
_ICLR.cc/2026/Conference — ICLR 2026 Poster_

### Official Review · Reviewer_JyfY · 2025-10-30

**Soundness:** 3
**Presentation:** 4
**Contribution:** 3
**Rating:** 2
**Confidence:** 4

**Summary:**

Proposes a method for inductive link prediction on knowledge hypergraphs (HKG). The approach closely follows ULTRA (for regular KGs), but modifies the representation of arity information in the relation graph (via an MLP position encoder) and replaces NBFNet (KGs) by HCNet (HGKs). Performs an experimental study on fully-inductive and node-inductive HKG link prediction. I like the paper, but I am not convinced that the experimental study is fair (see W2), and I am thus hesitant to recommend acceptance.

**Strengths:**

S1. Simple, convincing approach

S2. Relevant problem

S3. Experimental results are comprehensive and promising

**Weaknesses:**

W1. Low novelty. The proposed method is ultimately a rather straightforward modification of ULTRA. I do not count this against the paper too much though, because it is the natural approach and exploring and evaluating this approach provides value.

W2. Comparison to ULTRA not convincing. After reading the abstract of this paper, I immediately thought why not use ULTRA + reification. The authors then actually did this but in a way that is not convincing. To me, a key contribution of this paper is a fair and solid evaluation of the proposed method and alternatives, but I am not convinced that this has been done. This is for the following reasons:

1. The reification approach appears problematic. All relations become nodes, and only max-arity relations of form hasEntity-k(hyperedge, entity) + one hasRelationType relation are used. This severely limits the relation modelling capabilities of ULTRA. Instead, the natural reification is to relation-k(hyperedge, entity) and drop the hasRelationType relation. I feel that this approach needs to be explored, as it is what first comes to mind and as it appears much more promising.

2. ULTRA has not been retrained on the reified data (whereas the proposed method is). Since ULTRA has never seen reified relations, I'd not expect it to work well. A fair comparison would do the reification above and then train ULTRA on the same graphs (both KGs and HGs) as the proposed method. Only then I'd consider the study insightful and convincing.

W3. Comparison to HCNet not convincing. The authors seem to use random relation embeddings, as HCNet does not support new relations. I fail to see the value of this experiment; it clearly cannot lead to useful results. The experiment should either be dropped (ok with me) or perhaps (i) transductive results added and (ii) for the inductive setting, fold-in the new relations into HCNet.

**Questions:**

See W2.

---

> ### Author Response · Authors · 2025-11-19
>
> > “W1. Low novelty. The proposed method is ultimately a rather straightforward modification of ULTRA. I do not count this against the paper too much though, because it is the natural approach and exploring and evaluating this approach provides value.”
>
> While the overall design of HYPER has similarities with ULTRA, as the reviewer has acknowledged, exploring and evaluating this approach naturally provides value, and we respectfully argue that our contribution is **significant for the hypergraph domain**. The main contribution of our work is to develop the first foundation model for inference over arbitrary arity relations in knowledge hypergraphs, but the resulting model is not a naive modification of ULTRA. The core architectural design choice is the novel positional encoding scheme, which allows the model to understand the roles of entities within complex, variable-arity relations.  Since no foundation model for knowledge hypergraphs currently exists in the literature, we hope that HYPER can serve as a general blueprint for building such models in this domain.
>
> > “W2. 1. The reification approach appears problematic. All relations become nodes, and only max-arity relations of form hasEntity-k(hyperedge, entity) + one hasRelationType relation are used. This severely limits the relation modelling capabilities of ULTRA. Instead, the natural reification is to relation-k(hyperedge, entity) and drop the hasRelationType relation. I feel that this approach needs to be explored, as it is what first comes to mind and as it appears much more promising.
>
> Following the reviewer's advice, we have experimented with this alternative reification approach. We report the average zero-shot MRR below. Full experimental details are reported in **Appendix G.3** in the updated manuscripts:
>
> |Pretrain Mix|ULTRA (Original)|ULTRA (New)|HYPER|
> |:-|:-:|:-:|:-:|
> | 3KG | 0.138|0.161| **0.171**|
> | 4KG | 0.142|0.078| —|
> | 50KG | 0.147| 0.040| —|
> | 4HG | 0.125|0.168| **0.175**|
> | 3KG+2HG | 0.142| 0.183| **0.223**|
>
> Indeed, we observe that the proposed reification method lands between ULTRA and HYPER and is considerably better than the earlier reification method when the training graph contains similar data distributions (reified hypergraphs). However, the resulting performance still falls behind of HYPER’s performance. Moreover, we observe that ULTRA with the new reification generalizes worse to hypergraphs when pretrained over increasingly more KGs: ULTRA pretrained with 4KG and 50KG fails to produce competitive results. We hypothesize that such reification would create larger distribution shifts as it drastically increases the number of relations and does not lead to a holistic solution.
>
> > “W2. 2. ULTRA has not been retrained on the reified data (whereas the proposed method is). Since ULTRA has never seen reified relations, I'd not expect it to work well. A fair comparison would do the reification above and then train ULTRA on the same graphs (both KGs and HGs) as the proposed method. Only then I'd consider the study insightful and convincing.”
>
> We thank the reviewer for raising this point. We have added new experiments to **Tables 2 and 3** in the updated manuscript to ensure a fair comparison by pre-training the ULTRA baseline on the same hypergraph mixes (4HG and 3KG+2HG). The updated average zero-shot MRR results over 16 new datasets are:
>
> |**Model**|**4HG**|**3KG+2HG**|
> | :- | :-: | :-: |
> |ULTRA|0.125|0.166|
> |HYPER|**0.182**|**0.236**|
>
> These results further confirm the original trend: HYPER consistently outperforms ULTRA even when both are pre-trained on the same hypergraph data. This supports our conclusion that ULTRA's reification-based approach is less effective than HYPER's direct hypergraph modeling.

---

> ### Author Response · Authors · 2025-11-19
>
> > “W3. Comparison to HCNet not convincing. The authors seem to use random relation embeddings, as HCNet does not support new relations. I fail to see the value of this experiment; it clearly cannot lead to useful results. The experiment should either be dropped (ok with me) or perhaps (i) transductive results added and (ii) for the inductive setting, fold-in the new relations into HCNet.”
>
> We thank the reviewer for raising this concern. We have presented two main experiments in the paper:
> - **Node inductive + relation inductive** (**Table 2**). We have included the comparison with HCNet in this setup to showcase the limitation of existing methods when they observe increasingly novel relation types. We would like to highlight that this follows the footsteps of InGram [1], where the same experimental methology has been used in knowledge graphs. Nevertheless, this is not a core experiment in our paper and we agree with the reviewer that naively applying HCNet on increasingly novel relations (e.g., by assigning random relations) is necessarily going to fail.
> - **Node inductive (and relation transductive)** (**Table 3**). This experiment compares HYPER with HCNets in relation-transductive setup, where all relations are presented during training time to HCNets. This is precisely the comparison asked by the reviewer. Note that this is the classical node inductive setup HCNet is designed to handle, but even in this case, HYPER still has better performance, highlighting our methods’ strength. We are happy to highlight this experiment better if the presentation does not make this clear.
>
> [1] Lee J et al. InGram: Inductive knowledge graph embedding via relation graphs ICML 2023.

---

> ### Comment · Reviewer_JyfY · 2025-11-19
> **On response**
>
> Thanks for your response. Quick thoughts:
>
> On W1. We disagree on novelty of the method, but not on the value of your work. This is not a concern that will influence my recommendation, so I suggest to not discuss it further.
>
> On W2-1. The new experiments are helpful and match expectations. However, I am not happy with how this is currently handled. The main paper still has the flawed "bad" reification approach and the natural "good" one is pushed to the appendix; it should be the other way around. Moreover, in the table in your response, ULTRA does not seem to be pretrained on the reified data, which it should be (maybe as an additional column). Only then is the comparison fair.
>
> On W2-2. Which reification method has been used here?
>
> On W3.
>
> Tab 2: I stand to my assessment that the inclusion of HCNet with random embeddings in Tab 2 is not helpful. The fact that someone else already did this does not make this any more insightful.
>
> Tab 3 is fine, I may indeed have missed this. But "HYPER has better performance" is only partially true for end-to-end (i.e., not on JF-IND).

---

> > ### Author Response · Authors · 2025-11-20
> >
> > > “On W2-1. The new experiments are helpful and match expectations. However, I am not happy with how this is currently handled. The main paper still has the flawed "bad" reification approach and the natural "good" one is pushed to the appendix; it should be the other way around. Moreover, in the table in your response, ULTRA does not seem to be pretrained on the reified data, which it should be (maybe as an additional column). Only then is the comparison fair.”
> >
> > We thank the reviewer for their swift response. Following the reviewer’s advice, we have integrated the “natural” reification referred to by the reviewer in the updated paper and included the results for ULTRA under this “natural” reification to **Table 2** and **Table 3**. We present and discuss the earlier reification approach in **Appendix G.3**.
> >
> > Regarding the concern that ULTRA may not have been pretrained on the reified data: in our earlier response, the entries **ULTRA(4HG)** and **ULTRA(3KG+2HG)** indeed correspond to models pretrained on reified hypergraph data. Here, "HG" denotes that each hypergraph in the pretraining set was first reified using the new (natural) reification procedure, and the resulting knowledge graphs were then used to pretrain ULTRA. For example, **ULTRA(4HG)** is trained on the four hypergraphs after applying the new reification method. Thus, **ULTRA(4HG)** and **ULTRA(3KG+2HG)** are pretrained on the properly reified data, ensuring a fair comparison because the pretraining datasets are aligned across HYPER.
> >
> >
> > > “On W2-2. Which reification method has been used here?”
> >
> > These results refer to the old reification method used in the paper. Here, we present the updated table with the new reification method:
> >
> > | **Model** | **4HG** | **3KG+2HG** |
> > | :-------- | :-----: | :---------: |
> > | ULTRA    |  0.168  |    0.183    |
> > | HYPER     |  **0.182**  |    **0.236**    |
> >
> > > “G.3 states that the alternative reification approach "removes a two-hop detour". That's true, but it also allows ULTRA to use different representations for different relations; that's a key point to me.  **Appendix G.3** also states that the number of relations now "scale linearly with the number of relations originally in the knowledge hypergraph, which can be quite substantial". I fail to see why this is substantial; the same holds for HYPER. The number of relations in ULTRA compared to HYPER increases multiplicatively by the average arity (which, I'd guess, is typically small).”
> >
> > We agree with the reviewer on this point and rewrote **Appendix G.3** to accurately reflect that “the proposed reification methods allow ULTRA to use different representations for different relations”. We also removed the “scale linearly…” argument, and we agree that the number of relations in ULTRA will increase multiplicatively by the average arity under the new reification method.

---

> > > ### Comment · Reviewer_JyfY · 2025-11-20
> > >
> > > Thanks! I do not have time to look at your revision in the next couple of days, but if it indeed has all these changes, then I am going to recommend acceptance (nice work & great responses).
> > >
> > > (Not part of the review & no need to answer: why do you think HYPER outperforms ULTRA+reification?)

---

> > > > ### Author Response · Authors · 2025-11-21
> > > >
> > > > Many thanks for the thoughtful feedback and for your positive recommendation. We confirm that the current revision implements all the changes we described above, and we really appreciate your help in improving the paper.
> > > >
> > > > For the last question: we believe HYPER performs better due to its built-in inductive bias that cleanly separates node- and edge-level message passing with different operators, and works directly on hypergraph structure, while reification changes the relation embedding space and introduces distribution shifts that ULTRA must learn to handle.

---

> > > > > ### Comment · Reviewer_JyfY · 2025-11-21
> > > > >
> > > > > Thanks! I'd also add: (i) reified ULTRA separates entities of a fact by one additional hop, as you argued, (ii) the relation graph after reification only has t2t relationships between entities, (iii) ULTRA uses inverse relations for head prediction, which is not useful after reification (and thus shouldn't actually be used, including during pretraining).

---

> > > > > > ### Author Response · Authors · 2025-11-21
> > > > > >
> > > > > > Thank you for adding these additional points! We will include a detailed discussion in the updated manuscript.

---

> > > > > > > ### Comment · Reviewer_JyfY · 2025-11-24
> > > > > > > **A few more thoughts**
> > > > > > >
> > > > > > > I've looked at the paper; my two main concerns are now addressed and I recommend to accept the paper.
> > > > > > >
> > > > > > > A few remaining thoughts:
> > > > > > >
> > > > > > > Tab 2+3 are difficult to parse, a grouping by "pretraining mix" would make a direct comparison between HYPER and ULTRA less painful.
> > > > > > >
> > > > > > > Why is ULTRA (4KG) and ULTRA (50KG) reported, but not HYPER (4KG) and HYPER (50KG)?
> > > > > > >
> > > > > > > Likewise, why isn't ULTRA fine-tuned as well in Tabs 2+3?
> > > > > > >
> > > > > > > The main paper should spell out clearly that ULTRA is performing better on KG tasks.
> > > > > > >
> > > > > > > I find sentences such as this one rather fuzzy and misleading: "While reification technically enables the application of KGFMs to knowledge hypergraphs, it fails to capture the structure of entity-role interactions, resulting in significantly weaker performance." -- It clearly doesn't "loose" structure, as one can easily un-reify.

---

> > > > > > > > ### Author Response · Authors · 2025-11-24
> > > > > > > >
> > > > > > > > We thank the reviewer for their valuable feedback and for raising their score to an acceptance. Incorporating the feedback and additional experiments has substantially strengthened our paper. We will ensure that all remaining minor concerns are addressed in the final version.
> > > > > > > >
> > > > > > > > In response to the remarks, we will reorganize the rows by “pretraining mix” as suggested, to enhance clarity. We will also include the **HYPER(4KG/50KG)** and fine-tuned ULTRA experiments, which were previously excluded due to resource constraints during the rebuttal. We will explicitly state that “ULTRA generally performs better on standard Knowledge Graph tasks” with a direct reference to **Appendix E**. Additionally, we will revise the confusing reification sentence (regarding structure) to clarify that reification hinders KGFMs’ generalization due to inefficiencies induced by increased hop distances, ineffective inverse relations modeling, altered relation graph constructions, and deviation from standard pre-training distributions.

---

> > > > > > > > > ### Comment · Reviewer_JyfY · 2025-11-24
> > > > > > > > >
> > > > > > > > > Sounds good, thanks!

---

> > > > > > > > > > ### Author Response · Authors · 2025-11-28
> > > > > > > > > >
> > > > > > > > > > We thank the reviewer for raising their score from 2 to 8 following our rebuttal.

---

> ### Comment · Reviewer_JyfY · 2025-11-19
> **Afterthoughts on discussion in  G.3**
>
> G.3 states that the alternative reification approach "removes a two-hop detour". That's true, but it also allows ULTRA to use different representations for different relations; that's a key point to me.
>
> G.3 also states that the number of relations now "scale linearly with the number of relations originally in the knowledge hypergraph, which can be quite substantial". I fail to see why this is substantial; the same holds for HYPER. The number of relations in ULTRA compared HYPER increases multiplicatively by the average arity (which, I'd guess, is typically small).

---

### Official Review · Reviewer_6icB · 2025-10-31

**Soundness:** 3
**Presentation:** 3
**Contribution:** 3
**Rating:** 4
**Confidence:** 4

**Summary:**

This paper proposes a foundation model named HYPER for inductive link prediction. By constructing a relation graph G_rel and encoding relation representations on it, HYPER utilizes these learned representations to derive entity representations on the original knowledge hypergraph, enabling zero-shot generalization to knowledge hypergraphs of any arity, including new entities and new relations.

**Strengths:**

The paper proposes a framework that enables zero-shot generalization to knowledge hypergraphs of arbitrary arity, including novel nodes and relations, at test time.

The idea of encoding hyperedges using positional interaction is innovative.

The paper is clearly written with precise definitions, illustrative figures for concepts like relation graphs and reification, and provides code.

**Weaknesses:**

On L39, "The generality knowledge hypergraphs" should be corrected to "The generality of knowledge hypergraphs". The expression on L320-321 may also be problematic.

The pre-trained ULTRA model used for comparison, as described in Table 12, has not been pre-trained on any hypergraph, whereas HYPER(4HG) and HYPER(3KG+2HG) have both been trained on hypergraphs, which may lead to unfairness in the comparative experiment.

The idea of "constructing a relation graph $G_{rel}$ to learn relation embeddings and then using relation embeddings to learn entity embeddings" is similar to ULTRA, with only an extension of positional interaction encoding based on the concept of hyperedges, which may lack sufficient novelty.

**Questions:**

During the reification process, the positional information between nodes in a hyperedge is transformed into distinctions on edges of the form hasEntityi(edge id, ui). Does this lead to a loss of positional information?

HYPER's pre-training dataset is too small, potentially leading to overfitting to the relational structures of specific graphs. Could HYPER be trained with a larger pre-training dataset to assess its generalization ability?

---

> ### Author Response · Authors · 2025-11-19
>
> > W1. “On L39, "The generality knowledge hypergraphs" should be corrected to "The generality of knowledge hypergraphs". The expression on L320-321 may also be problematic.”
>
> We thank the reviewer for pointing out the typos and misformulations. We have fixed these minor issues in the updated paper.
>
> > W2. “The pre-trained ULTRA model used for comparison, as described in Table 12, has not been pre-trained on any hypergraph, whereas HYPER(4HG) and HYPER(3KG+2HG) have both been trained on hypergraphs, which may lead to unfairness in the comparative experiment.”
>
> We thank the reviewer for raising this point. To address their concern, we have added new experiments to **Tables 2 and 3** in the updated manuscript to ensure a fair comparison by pre-training the ULTRA baseline on the same (but reified) hypergraph mixes (4HG and 3KG+2HG). The updated average zero-shot MRR results over 16 new datasets are:
>
> | **Model** | **4HG** | **3KG+2HG** |
> | :-------- | :-----: | :---------: |
> | ULTRA    |  0.125  |    0.166    |
> | HYPER     |  **0.182**  |    **0.236**    |
>
> These results further confirm the original trend: HYPER consistently outperforms ULTRA even when both are pre-trained on the same hypergraph data. This supports our conclusion that ULTRA's reification-based approach is much less effective than HYPER's direct hypergraph modeling.
>
> > W3. “The idea of "constructing a relation graph $G_rel$ to learn relation embeddings and then using relation embeddings to learn entity embeddings" is similar to ULTRA, with only an extension of positional interaction encoding based on the concept of hyperedges, which may lack sufficient novelty.”
>
> While the overall design of HYPER has similarities with ULTRA, exploring and evaluating this approach naturally provides value, and we respectfully argue that our contribution is **significant for the hypergraph domain**. The main contribution of our work is to develop the first foundation model for inference over arbitrary arity relations in knowledge hypergraphs, but the resulting model is not a naive modification of ULTRA. As pointed out by the reviewer, the core architectural design choice is the novel positional encoding scheme, which allows the model to understand the roles of entities within complex, variable-arity relations. Since no foundation model for knowledge hypergraphs currently exists in the literature, we hope that HYPER can serve as a general blueprint for building such models in this domain.
>
> > Q1. “During the reification process, the positional information between nodes in a hyperedge is transformed into distinctions on edges of the form $hasEntityi(edge id, ui)$. Does this lead to a loss of positional information?”
>
> This will not **lose** the positional information. We show this by proving that the reification process is invertible: Given a reified knowledge hypergraph, one can reconstruct the input knowledge hypergraph by first identifying 3 disjoint sets of nodes: relation node, edge ID node, and original entities. Relation nodes can be recovered as they are the exact nodes that have incoming *hasRelationType* edges. We can also recover the edgeID node by tracing the source of all *hasRelationType* edges. Then, for each edgeID node $\mathrm{edge}\_\mathrm{id}$, we trace all its neighbors, and construct $r(u_1,...,u_k)$ if $\mathrm{hasEntity}_i(\mathrm{edge}\_\mathrm{id}, u_i), \mathrm{hasRelationType}(\mathrm{edge}\_\mathrm{id}, r)$ exist. Thus, all the information is preserved during transformation, including the positional information, which is stored as different edge types in the reified knowledge graphs.
>
>
>
> > Q2. “HYPER's pre-training dataset is too small, potentially leading to overfitting to the relational structures of specific graphs. Could HYPER be trained with a larger pre-training dataset to assess its generalization ability?”
>
> We agree with the reviewer that the pre-training datasets are not very diverse, but this due to the fact that the number of knowledge hypergraph datasets currently available is limited. Following reviewer suggestions, we have trained a HYPER instance with a larger pretraining set (3KG+4HG), which incorporates all 3 knowledge graphs and 4 hypergraphs.  The average zero-shot MRR results over 16 new datasets are:
>
> | **Pretrain** | **3KG+2HG** | **3KG+4HG** |
> | :-------- | :-----: | :---------: |
> | HYPER     |  0.236  |    **0.241**    |
>
> This result supports the hypothesis that pre-training on a larger and more diverse mix of relational structures is beneficial and leads to improved generalization.

---

> ### Comment · Reviewer_6icB · 2025-11-26
> **update after rebuttal**
>
> Thanks for the authors' rebuttal. The author addressed my concerns, and I will keep my score.

---

> > ### Author Response · Authors · 2025-11-26
> >
> > We thank the reviewer for their response and for confirming that all their concerns are addressed. We kindly ask the reviewer to consider revisiting their score given that the rebuttal addressed their concerns. We strongly believe the reviews helped us improve the paper.

---

### Official Review · Reviewer_kmsQ · 2025-11-03

**Soundness:** 2
**Presentation:** 2
**Contribution:** 3
**Rating:** 6
**Confidence:** 3

**Summary:**

This paper introduces a foundation model (HYPER) for predicting missing hyperedges in knowledge hypergraphs that contain unseen entities and unseen relation types. HYPER could generalize to arbitrary-arity relations through a relation graph encoder that captures positional interactions between relations. The paper also introduces 16 new inductive benchmark datasets derived from existing knowledge hypergraphs.

**Strengths:**

1.	HYPER is a general framework that can transfer learned relational patterns across different relation types and arities. This is the first foundation model that supports zero-shot generalization on knowledge hypergraphs of arbitrary arity.
2.	16 new inductive benchmark datasets derived from existing knowledge hypergraphs are constructed for evaluation.
3. Empirical investigation over the positional interaction encoding scheme

**Weaknesses:**

1.	The positional interaction encoding $\mathrm{EncPI}((a,b)) = \mathrm{MLP}([p_a \| p_b])$ may violate symmetry and lacks proof of equivariance or smooth extrapolation to unseen arities; this weakens the theoretical basis for HYPER’s claimed generalization.

2.	While HYPER demonstrates strong zero-shot generalization across diverse hypergraph benchmarks, how practical is it for large-scale real-world knowledge systems—given that the number of positional interactions grows quadratically with relation arity and may impose significant computational and memory costs during training and inference?

**Questions:**

Refer to the Weakness section.

---

> ### Author Response · Authors · 2025-11-19
>
> > “W1. The positional interaction encoding $EncPI((a,b)) = MLP([p_a\|p_b])$ may violate symmetry and lacks proof of equivariance or smooth extrapolation to unseen arities; this weakens the theoretical basis for HYPER’s claimed generalization.”
>
> Following the reviewer’s comment, we have included additional theorems and proofs in **Appendix C** of the updated manuscript. We have also updated the theoretical statements to formally show the following:
> 1)  HYPER with $\mathrm{Enc}_{\text{PI}}(a,b)=\mathrm{MLP}([\mathbf{p}_a \| \mathbf{p}_b])$ computes an invariant over nodes and relations in knowledge hypergraphs, and is thus equivariant under renaming of nodes or relations. (**Proposition C.1, Appendix C.2**)
> 2) For any input knowledge hypergraph there exists a parameterization of HYPER such that the $\mathrm{Enc}_{\text{PI}}(a,b)$ is injective.  (**Theorem C.2.1, Appendix C.3**)
> 3)  $\mathrm{Enc}_{\text{PI}}(a,b)$ has its images contained in a compact set. (**Theorem C.2.2, Appendix C.3**)
> 4) $\mathrm{Enc}_{\text{PI}}(a,b)$ is lipschitz and thus smooth. (**Theorem C.2.3, Appendix C.3**)
>
> Specifically, 3 shows that for unseen arity indices $(a,b)$, the resulting positional representations remain within the same compact set in Euclidean space (thus bounded) as those observed during training.
>
>
> > “W2. While HYPER demonstrates strong zero-shot generalization across diverse hypergraph benchmarks, how practical is it for large-scale real-world knowledge systems—given that the number of positional interactions grows quadratically with relation arity and may impose significant computational and memory costs during training and inference?”
>
> We thank the reviewer for acknowledging the strong zero-shot generalization across diverse hypergraph benchmarks. While the maximum arity of the tested benchmark is $22$, the positional interactions grow quadratically with relation arity – which we acknowledged in the limitations part of our original submission.  That being said, we would like to emphasize that our approach for adding positional encodings scales linearly in the number of edges $E$. Observe that this is efficient for most real-world graphs, as they are typically composed of relations with a small arity (i.e., arity bounded by a small constant $K$). In this sense, our approach avoids storing a quadratic number of positional interactions, and instead relies on learning a unified encoder that maps these positional interactions into a shared space, which can be done efficiently for most real-world graphs.

---

### Official Review · Reviewer_kwmy · 2025-11-03

**Soundness:** 3
**Presentation:** 4
**Contribution:** 3
**Rating:** 8
**Confidence:** 4

**Summary:**

The paper proposes and architecture for inductive link prediction with unseen entities as well as unseen relations over knowledge hypergraphs  via knowledge hypergraph foundational models. The authors show the feasibility of the approach over link prediction task with standard evaluation metrics and propose a set of new datasets.

**Strengths:**

- The paper introduces a problem which is very much mainstream in the field of knowledge hypergraphs and address an important gap.
- The paper is very well written and easy to read.
- The authors performed thorough experimentation with the comparative results along with the analysis.

**Weaknesses:**

- the authors could cite another relevant paper which is performing inductive link prediction over knowledge graphs considering features related to the relations [1].
- The role of encoding positional encoding could be explained with the help of an example to show its importance for the proposed approach.
- It so far seems like a combination of approaches, the authors should highlight the main theoretical contribution in the paper.
- Table 3 show the MRR results where HCNET outperforms HYPER. Also in many cases the improvements are only marginal. Can authors shed the light over these kind of results since there can be a possibility that the results of the baselines might improve with extensive hyper-parameter optimization.



[1] https://dl.acm.org/doi/10.1145/3579051.3579066

**Questions:**

See the weaknesses of the approach.

---

> ### Author Response · Authors · 2025-11-19
>
> > “W1. The authors could cite another relevant paper which is performing inductive link prediction over knowledge graphs considering features related to the relations [1].”
>
> We thank the reviewer for pointing out the relevant literature. Following the reviewer's advice, we have updated our related work to include RAILD as one of the models for inductive link prediction over knowledge graphs with new relations.
>
> > “W2. The role of encoding positional encoding could be explained with the help of an example to show its importance for the proposed approach.”
>
> We have presented a concrete example in **Figure 2** and **Figure 3**. Let us elaborate this point further on this concrete example:
> 1. When the same entity appears at argument position $i$ of relation $r_1$ and position $j$ of $r_2$, we add a directed edge ($r_1, r_2$) labeled $(i,j)$ in the relation-interaction graph. For example, **Montreal** appears at position $2$ in the relation **AtConference** and at position $3$ in the relation **Research** and so we add a directed edge between **AtConference** and **Research** labeled with $(2,3)$.
> 2. Each positional pair (a,b) is then turned into a typed message using the encoding: $\mathrm{Enc}_{\text{PI}}(a,b)=\mathrm{MLP}([\mathbf{p}_a \| \mathbf{p}_b])$. This helps the model learn the correspondences across relations, i.e., the implicit node types (role) that these relations share at the corresponding position. Without these messages, $(2,3)$ would be indistinguishable from $(1,1)$ or $(3,2)$, collapsing the role semantics and harming transferrability.
>
> This is validated in our experiment in **Section 5.5**, where we randomly and inconsistently permute argument positions and the performance drops dramatically as a result. This ablation shows that the positional information is essential for HYPER’s generalization.
>
> > “W3. It so far seems like a combination of approaches, the authors should highlight the main theoretical contribution in the paper.”
>
> To provide additional insight, we added theoretical results in **Appendix C**:
> 1. We formalize the notion of invariants for link prediction with knowledge hypergraphs (**Appendix C.1**) and prove that HYPER computes such an invariant (**Proposition C.1, Appendix C.2**)
> 2. We prove key properties of the positional-interaction encoder $\mathrm{Enc}_{\text{PI}}$ (**Theorem C.2, Appendix C.3**):
>     - $\mathrm{Enc}_{\text{PI}}$ has an image contained in a compact subset. Thus, even for unseen arity indices $(a, b)$, the resulting positional representations remain within the same bounded set as those observed during training.
>     - For any input knowledge hypergraph there exists a parameterization of HYPER such that the $\mathrm{Enc}_{\text{PI}}$ is injective.
>     - $\mathrm{Enc}_{\text{PI}}$ is Lipschitz and thus smooth.
>
> Following the reviewer’s advice, we have also highlighted the theoretical contributions in the main body of the paper.
>
>
> > “W4. Table 3 show the MRR results where HCNET outperforms HYPER. Also in many cases the improvements are only marginal. Can authors shed the light over these kind of results since there can be a possibility that the results of the baselines might improve with extensive hyper-parameter optimization.”
>
> We would like to emphasize that the purpose of the experiment reported in **Table 3** is different from the others, as it compares end-to-end methods (which observed all relation types) to HYPER (which did not observe the relation types). The surprising finding is that HYPER can still outperform end-to-end methods, despite operating in a zero-shot inference setting. This shows the promise of developing a foundation model: instead of extensively hyperparameter-tuning transductive methods on specific datasets, one can instead pretrain a foundation model like HYPER and directly apply to unseen data – obtaining on-par or even better results when compared to specialized end-to-end counterparts.

---

### Author Response · Authors · 2025-11-30
**Summary of rebuttal and changes**

Dear AC,

We wish to express our sincere gratitude for the additional time and effort you are dedicating to evaluating our work under these challenging circumstances. To facilitate your assessment, we provide below a concise overview of the main changes introduced during the rebuttal:

- **Reviewer kwmy (Score 8)**: Added the theoretical analysis to formally show that HYPER with the proposed EncPI computes invariants over nodes and relations, has range within a compact set, and is Lipschitz.


- **Reviewer kmsQ (Score 6)**: Clarified how the proposed positional–interaction encoder computes invariants theoretically, and how the newly derived theoretical guarantees (compactness, Lipschitz, injectivity) support the empirical behavior of HYPER.


- **Reviewer 6icB (Score 4)**: Strengthened the experimental comparison between ULTRA and HYPER using the new ULTRA(4HG) and ULTRA(3KG+2HG) baselines pretrained on reified hypergraphs; added a more diverse pretraining mix by including additional checkpoints over HYPER(3KG+4HG) to show the generalizability of HYPER; demonstrated our reification method does not lose positional information. The reviewer replied with *“The author addressed my concerns"*, indicating their satisfaction with our response.

- **Reviewer JyfY (Score 2 → 8)**: Added ULTRA baselines pretrained on reified knowledge hypergraphs, ULTRA(4HG) and ULTRA(3KG+2HG), to ensure a fair comparison between ULTRA and HYPER; incorporated additional reification techniques suggested by the reviewer into the main experiments and Appendix G; clarified our contributions and addressed the initial concerns in detail, leading to a score increase from 2 to 8. The reviewer responded to the rebuttal with the remark *“nice work & great responses,”* indicating their overall satisfaction.


These changes led to **scores of 8, 8, 6, and 4**, corresponding to an **average of 6.5**, *prior* to the review period being frozen. All reviewers recognized that our paper addresses a timely and relevant problem in knowledge hypergraphs and proposes a simple yet convincing framework (HYPER) for zero-shot generalization to arbitrary arity via an innovative positional–interaction encoder.

We trust that this summary will be beneficial to your assessment.

Kind regards,

Authors of paper 13425

---

### Meta-Review · Area_Chair_4LSN · 2026-01-06

**Summary:**

The reviewers agree that this paper makes a strong and timely contribution by introducing HYPER, the first foundation model designed for inductive link prediction over knowledge hypergraphs with both unseen entities and unseen relation types, addressing an important and previously unmet need in the literature. The proposed positional–interaction encoder enables principled generalization across arbitrary relation arities, and the paper is further strengthened by the construction of 16 new inductive benchmarks and extensive empirical validation across node-only and node-and-relation inductive settings. Reviewers found the approach conceptually clean, well-motivated, and empirically convincing, with clear performance gains over strong baselines under fair and carefully controlled comparisons.

**Reviewer Concerns:**

Initial concerns regarding novelty relative to ULTRA, fairness of reification-based baselines, theoretical guarantees of the positional encoding, scalability, and clarity of experimental design were comprehensively addressed in the rebuttal through substantial new experiments, improved baseline pretraining, alternative reification strategies, added theoretical analysis establishing invariance and smoothness properties, and clearer presentation in both the main paper and appendices.

**Reviewer Scores:**

After discussion and rebuttal, reviewer sentiment converged strongly toward acceptance, including a notable score increase from initially negative to clearly positive, with reviewers explicitly stating that their main concerns had been resolved and recommending acceptance based on the paper’s strengthened rigor, clarity, and impact.

---

### Decision · Program_Chairs · 2026-01-26

Accept (Poster)